# Lipid-Based Nanotechnologies for Delivery of Green Tea Catechins: Advances, Challenges, and Therapeutic Potential

**DOI:** 10.3390/pharmaceutics17080985

**Published:** 2025-07-30

**Authors:** Stanila Stoeva-Grigorova, Nadezhda Ivanova, Yoana Sotirova, Maya Radeva-Ilieva, Nadezhda Hvarchanova, Kaloyan Georgiev

**Affiliations:** 1Department of Pharmacology, Toxicology and Pharmacotherapy, Faculty of Pharmacy, Medical University of Varna, 84 “Tsar Osvoboditel” Blvd., 9000 Varna, Bulgaria; maya.radeva@mu-varna.bg (M.R.-I.); nadejda.hvarchanova@mu-varna.bg (N.H.); kaloyan.georgiev@mu-varna.bg (K.G.); 2Department of Pharmaceutical Technologies, Faculty of Pharmacy, Medical University of Varna, 84 “Tsar Osvoboditel” Blvd., 9000 Varna, Bulgaria; nadejda.ivanova@mu-varna.bg (N.I.); yoana.sotirova@mu-varna.bg (Y.S.)

**Keywords:** bioavailability enhancement, drug delivery systems, epigallocatechin gallate, flavonoid antioxidants, functional nutrition, green tea catechins, lipid-based nanocarriers, phytochemical therapeutics, phytotherapy

## Abstract

Knowing the superior biochemical defense mechanisms of sessile organisms, it is not hard to believe the cure for any human sickness might be hidden in nature—we “just” have to identify it and make it safely available in the right dose to our organs and cells that are in need. For decades, green tea catechins (GTCs) have been a case in point. Because of their low redox potential and favorable positioning of hydroxyl groups, these flavonoid representatives (namely, catechin—C, epicatechin—EC, epicatechin gallate—ECG, epigallocatechin—EGC, epigallocatechin gallate—EGCG) are among the most potent plant-derived (and not only) antioxidants. The proven anti-inflammatory, neuroprotective, antimicrobial, and anticarcinogenic properties of these phytochemicals further contribute to their favorable pharmacological profile. Doubtlessly, GTCs hold the potential to “cope” with the majority of today‘s socially significant diseases, yet their mass use in clinical practice is still limited. Several factors related to the compounds’ membrane penetrability, chemical stability, and solubility overall determine their low bioavailability. Moreover, the antioxidant-to-pro-oxidant transitioning behavior of GTCs is highly conditional and, to a certain degree, unpredictable. The nanoparticulate delivery systems represent a logical approach to overcoming one or more of these therapeutic challenges. This review particularly focuses on the lipid-based nanotechnologies known to be a leading choice when it comes to drug permeation enhancement and not drug release modification nor drug stabilization solely. It is our goal to present the privileges of encapsulating green tea catechins in either vesicular or particulate lipid carriers with respect to the increasingly popular trends of advanced phytotherapy and functional nutrition.

## 1. Introduction

Plant-derived natural products play a pivotal role in the pharmaceutical and biotechnological industries. Despite advances in modern technologies, they continue to be utilized for therapeutic purposes and remain a major source for the discovery and development of novel drug candidates [1,2,3]. In 2023, during the First Global Summit on Traditional Medicine convened by the World Health Organization, it was officially reported that traditional medicine is practiced in 88% of countries worldwide, recognizing that for millions of people, this approach represents the initial step in their healthcare journey [4]. The market for plant-derived medicinal substances encompasses a broad spectrum of products within the pharmaceutical, nutraceutical, and cosmetic industries, underscoring their considerable potential to enhance human health and well-being [5]. This sector has experienced sustained growth, fueled by the increasing incorporation of herbal therapies into mainstream medicine and rising global demand for natural health products. Valued at USD 164.6 billion in 2021, the market is projected to grow at a compound annual growth rate of 7.8% through 2028 [6].

*Camellia sinensis* (L.) Kuntze is primarily known as the raw material for green tea, the world’s second most consumed beverage after water [7]. According to the World Green Tea Association, approximately 600,000 tons are consumed annually worldwide, accounting for about one-fifth of total tea consumption [8]. Ongoing phytochemical research has identified over 500 chemical constituents in the plant leaves, including alkaloids, polyphenols, amino acids, aromatic compounds, carbohydrates, organic acids, minerals, vitamins, enzymes, pigments, and other bioactive compounds [5,7,9,10,11]. Extensive in vitro and in vivo studies have established that green tea possesses antioxidant, anti-inflammatory, neuroprotective, antimicrobial, anticancer, and cancer-preventive properties, as well as anti-aging effects and benefits for weight management [7,12,13,14,15,16,17]. These biological activities are largely attributed to its polyphenolic compounds, particularly catechins. Among the predominant catechins in green tea leaves are (−)-epigallocatechin-3-gallate (EGCG), (−)-epicatechin-3-gallate (ECG), (−)-epigallocatechin (EGC), and (−)-epicatechin (EC), with EGCG being the most abundant and recognized as the primary bioactive constituent responsible for the majority of its therapeutic effects [17].

Increasing interest in natural compounds for managing chronic diseases has intensified scientific focus on catechins, prompting substantial efforts to standardize their consumption [17,18,19]. However, their clinical application remains considerably limited, primarily due to inadequate physicochemical stability and low oral bioavailability. To address these biopharmaceutical limitations, a range of strategies is under active investigation, including:Co-administration with complementary bioactive agents;Structural modification;Encapsulation into nanoscaled drug delivery systems [20].

The latter category is, without doubt, of greatest interest to contemporary research and offers a vast and diverse toolkit for pharmaceutical development. Based on the size criteria only, pharmaceutical nanotechnologies unite versatile materials and compositions that may contribute with highly specific advantages and opportunities. In this regard, polymeric-based nanomaterials are strong in ensuring modified drug release, providing high encapsulation capacity, inertness, and drug stabilization but suffer low reproducibility and toxicity concerns [21]. Inorganic nanoparticles normally represent chemically stable systems with easily reproducible manufacturing and, in particular cases (e.g., metal nanoparticles) even their own pharmacological activity, but they hold the risk of adverse reactions and bio-incompatibility [22]. The sub-category of lipid-based nanotechnologies, along with hybrid nanotechnologies, stand out as the ones of choice when the need for enhanced permeation exists (regardless of the application site) and high biocompatibility for the various routes of drug administration is sought after. Additionally, lipid-based nanocarriers (including vesicular systems and lipid nanoparticles (LNs)) possess the ability for drug protection. Major challenges related to them are chemical preservation, physical endurance, low encapsulation capacity, the hard-to-reproduce methods for preparation, and most and all—the high production cost that comes with all these disadvantages. Still, these technologies evolve very fast, and step-by-step overcome some limitations. Lipid-based nanoparticle encapsulation has shown particular promise for the delivery of catechins, offering improved stability, enhanced transmucosal or (trans)cutaneous absorption, and greater systemic or local bioavailability [23,24]. Experimental studies have demonstrated that such delivery platforms significantly augment the in vivo efficacy of catechins, with especially notable enhancements observed for EGCG [17].

This review is primarily intended for pharmaceutical scientists and nanotechnology researchers engaged in the development of oral drug delivery systems, with a particular focus on lipid-based carriers aimed at enhancing the bioavailability and, consequently, the therapeutic efficacy of catechins. We believe that the content may also be of interest to clinicians and food technologists concerned with the application of this class of phenolic compounds.

## 2. Methods

To fulfill the research objective, we collected and critically evaluated contemporary scientific evidence regarding the benefits of encapsulating green tea catechins (GTCs) within vesicular and particulate lipid-based carriers, with emphasis on the evolving paradigms of advanced phytotherapy and functional nutrition. To gain a comprehensive understanding of the topic, we conducted an extensive literature search using several leading scientific databases, including Web of Science, Scopus, PubMed, and ResearchGate. While our primary focus was on studies published within the last decade, the search encompassed literature spanning from 1943 to 2025, retrieving over 290 relevant sources.


**Inclusion Criteria:**


Original research articles, reviews, and preclinical studies focused on lipid-based nanocarriers (including but not limited to: *lipid-based nanoparticles*, *solid lipid nanoparticles*, *nanostructured lipid carriers*, *liposomes*, *phytosomes*, *herbosomes*, *nanoemulsions*, *microemulsions*, *cubosomes*, *ethosomes*, *transferosomes*, *chitosomes*, *self-emulsifying drug delivery systems*, *invasomes*, and *niosomes*) for the delivery of GTCs;Articles reporting physicochemical characterization, stability, bioavailability, pharmacokinetics, pharmacodynamics, and therapeutic efficacy of GTC-loaded lipid nanocarriers;Both in vitro and in vivo studies assessing the biological effects, toxicity, and delivery efficiency of lipid-based GTC formulations;Studies published in English or accompanied by an accessible English abstract;Research addressing health-related outcomes, including antimicrobial, antioxidant, anticancer, anti-inflammatory, and metabolic activities.


**Exclusion Criteria:**


Studies focusing on delivery systems that are not lipid-based or nanocarriers unrelated to lipid nanotechnology;Articles lacking experimental data (e.g., opinions, editorials), unless providing critical review or analysis;Studies involving catechins derived from sources other than green tea, unless directly comparable;Reports insufficiently detailed in methodology to allow reproducibility or data verification;Publications without available translations.Research exclusively centered on synthetic analogs or derivatives of catechins without direct relevance to natural green tea catechins.

## 3. The Nature of Catechins

As secondary plant metabolites characterized by multiple phenolic hydroxyl groups in their structure, the catechins fall within the vast category of biologically active compounds (BACs) known as polyphenols. More specifically, they represent a unique class of flavonoids united by a flavan-3-ol configuration [25,26]. The first identified representative—catechin, carries the name of the original source used for its isolation—*Acacia catechu* (L.f.) Willd [27,28]. Today, there are recognized five major members of the catechin family, namely C, EC, EGC, ECG, and EGCG (Figure 1) [29,30].

### 3.1. Natural Sources of Catechins

The primary source of catechins is the tea plant foliage, particularly the variety of *green* teas being yielded from its harvesting and processing—e.g., Sencha, Matcha, Gyokuro, Bancha, etc. [31,32,33,34,35]. In fact, *white* teas, although not in that mass use, compete with *green* teas for the highest total catechin content [35,36,37,38]; contrariwise, *Pu-erh* teas, *black* teas, *oolong* teas, and other types of teas obtained through fermentation generally preserve considerably lower amounts of monomeric catechins and are more abundant of their dimeric and polymeric derivatives (e.g., theaflavins, thearubigins) (Figure 1) [38,39,40,41]. Additionally, various factors affect green tea composition such as geographical location, cultivation conditions, soil, climate, temperature as well as tea leaves processing. Furthermore, different types of green tea are available such as Chinese, Japanese, Korean, and others based on their geographical origin resulting in different qualitative and quantitative content of bioactive compounds as well as organoleptic properties [17]. It should be noted that soil is also crucial for green tea growth and composition. Optimal conditions include sandy loam or sandy clay soil enriched with humus that content above 2% of organic matter, pH in the range of 4.0–5.5 and a good drainage. Interestingly, the application of ammonium-based fertilizers may either increase or decrease the polyphenol content [17].

Alternative natural sources of catechins in the human diet are red wine, black grapes, strawberries, apricots, apples, blackberries, broad beans, cherries, pears, raspberries, and cacao. However, their biomass is rarely implemented for the extraction of these BACs for scientific or medicinal purposes, wherefore the GTCs are prioritized in the next sub-sections of this paper [42]. Irrespective of the origin, the gallic acid-esterified catechins (galloylated catechins) are predominant, with EGCG being the most prevalent in the majority of herbal substances and most pharmacologically potent [42,43].

### 3.2. Biosynthesis

The catechins are ultimate products of the flavonoid biosynthetic pathway and precursors of the dimeric, oligomeric, and polymeric polyphenols, such as the theaflavins, thearubigins, and condensed tannins (proanthocyanidins) (Figure 1) [44,45,46]. The direct forerunners of the *trans*-conformational catechins—(+)-catechin (C) and (+)-gallocatechin—are the leucoanthocyanidins (leucocyanidin and leucodelphinidin, respectively), whereas the *cis*-flavan-3-ols-(−)-EC and (−)-EGC, originate from the reduction in anthocyanidins (cyanidine and delphinidin, respectively). These latest biotransformations are determined by the action of the leucoanthocyanidin reductase and the anthocyanidin reductase [45,46,47,48]. The galloylated catechins—ECG and EGCG—are formed in the presence of 1-O-β-glucogallin upon the stepping into action of serine carboxypeptidase-like acyltransferases [49,50].

### 3.3. Extraction and Processing

The herbal substances of choice for the isolation of catechins are *green* teas, for they are accessible and rich (in up to 80 mg/g total flavan-3-ols) sources [42,51]. Among them, the Sencha and Matcha types are ranked first for total catechin content [52,53,54]. Alternatively, powdered green tea extracts standardized to a minimum content of polyphenols, catechins, and/or EGCG might be preferred for higher yield [32]. In any case, the storage conditions of the herbal substance should be so maintained that any negative influence on the catechin content (caused by moisture, light, temperature, or adjuvants) be avoided [55,56].

The extraction and purification of catechins meet several challenges, among which their inherent chemical instability stands out. The natural catechins are susceptible to oxidation, epimerization, and polymerization under the influence of temperature, light, alkaline pH, and some ions [42,56,57]. This fundamental knowledge of them should be considered in any step of their processing, including further nano-formulation. The standard solid–liquid extraction (SLE) techniques for GTCs maneuver between the compounds’ chemical preservation, sufficient and selective solubilization, compulsory decaffeination, and successful purification [42,57]. Suitable extractants in the early steps of extraction are polar solvents or mixtures in which the GTCs dissolve well, e.g., water, ethanol, and methanol [57,58,59]. The processing of the primary extract with non-polar or semi-polar solvents, such as dichloromethane or chloroform (known as non-solvents for polyphenols), is useful in decaffeination [60,61,62,63]. Following separation/purification with ethyl acetate or propyl acetate is known to allow the selective isolation of catechins [57,64,65,66]. Conventional methods like infusion, maceration, or Soxhlet extraction are applicable in the so-described general methodology for GTCs extraction, whereas preparative chromatography or solid phase extraction might be carried out for obtaining individual catechin fractions with higher purity (>80%) [42,57]. As higher temperatures and alkaline pH are generally required during the SLE of GTCs for ensuring/improving the compounds’ dissolution and diffusivity, it is a common goal to minimize the extraction time under the stability-unfavorable conditions by improving the extraction efficiency. Techniques upgraded by microwaves, ultrasound, or pressurized liquids, for example, are often implemented in this respect [42]. Superior to the classic SLE is the supercritical CO_2_ extraction (SC-CO_2_)—an eco-friendly and high-yield extraction technology for labile BACs performed with the aid of fluidized CO_2_ gas. For the purposes of GTCs supercritical CO_2_ extraction, the addition of polar solvents, such as ethanol or methanol, is required [67,68]. Besides the SC-CO_2_, the application of ionic liquids (liquid-state salts) or deep eutectic solvents (eutectic solvents composed of two or more primary metabolites, i.e., organic acids, sugars, alcohols, amino acids) have shown promising perspectives for catechins’ isolation in recent studies [69,70].

The choice between using pure catechin reagents or extracts derived directly from tea leaves ultimately depends on the specific objectives of the study. On the one hand, tea, as a complex natural matrix, provides a rich and diverse mixture of catechins in conjunction with other bioactive compounds (e.g., caffeine, theanine, and various polyphenols), which may act synergistically to potentiate overall biological effects [13,17,33]. For large-scale or preliminary investigations, extracting catechins from tea leaves may prove more cost-effective than purchasing high-purity individual compounds—particularly when the aim is to study real-world formulations or dietary sources. This approach also more accurately reflects the manner in which catechins are consumed in the human diet, thereby rendering the results more physiologically relevant for nutritional or therapeutic applications [71]. Furthermore, it allows for the examination of how cultivar type and/or processing method influence biological activity [17,72,73]. On the other hand, the use of purified catechins enables the administration of a precisely controlled and defined dose, eliminating the variability introduced by other constituents present in tea extracts. Pure reagents are particularly suited for studies focused on specific molecular mechanisms or distinct pharmacological effects, as they allow for more unambiguous interpretation of results. In conclusion, tea extracts are preferable for investigations of complex natural mixtures and real-life applications, whereas pure catechin reagents are more appropriate for mechanistic and/or dose-specific studies [74,75].

### 3.4. Physicochemical Properties

In a purified state, the GTCs appear as colorless crystals with a bitter and astringent taste, more pronounced by the galloylated derivatives [42]. To a great extent, the in vitro and in vivo behavior of catechins in all aspects (e.g., solubility, extraction efficiency, stability, complexation, absorption, biotransformation, antioxidant activity) is determined by the number and positioning of the hydroxyl groups in their structure (counting 5 to 8), and more specifically, by the number and strength of intra- and intermolecular hydrogen bonds occurring [30,76,77]. Another factor that differentiates the representatives in this class is the molecular mass. Table 1 summarizes available data for H-binding capacity, molecular weight, and solubility of GTCs.

When in a polar medium, the GTCs form a bulky hydration shell as a result of multiple hydrogen bonds. This phenomenon usually eases their molecular dissolution but hinders their diffusivity and thus ability to be extracted, absorbed, etc. [30,82,83]. Moreover, the so-increased hydrodynamic diameter and molecular weight may potentiate desired or undesired colloidal instability. The latter is particularly valid in the presence of other polar (macro)molecules, sometimes purposely used for precipitation (creaming-down) of catechins in vitro (e.g., polyvinylpyrrolidone, cellulose derivatives) or nanoformulation, other times, a part of the normal molecular surrounding in the gut, plasma, etc. (e.g., proteins, amino acids, sugars) [83,84,85,86,87]. Indeed, the relatively new trend for isolation of catechins with the aid of natural deep eutectic solvents is settled on this foundation [69].

The GTCs are unstable compounds and easily step into oxidation and polymerization types of chemical interactions in vitro and in vivo. Thereby, they form catechin derivatives of higher molecular order (theaflavins, thearubigins, and proanthocyanidins), which are characterized by weaker antioxidant properties and lower bioavailability [88,89,90]. The chemical instability of the individual catechins increases as their ability to donate protons increases in an ionized state (at alkaline pH) [91]. The major structural particularity dictating the decomposition rate, as mentioned above, is the count and positioning of hydroxyl groups. The galloylated catechins—ECG and EGCG—are more prone to oxidation, for their structural particularities define lower redox potential and an additional autocatalytic mechanism of degradation [76,92]. Furthermore, the catechins are among the rare flavonoids that exist mostly in an aglycone form (as non-glycosides), and by that, their susceptibility to oxidation is also enhanced [93,94]. Relative stabilization of the GTCs could be achieved at pH around 4 and low temperatures as long as it is considered that these conditions may reduce the antioxidant potency of the compounds (and any pharmacological activity thereupon) [91,95]. The more advanced strategies for stabilization and bioavailability improvement of catechins rely on molecular and supramolecular modifications (e.g., by complexation, coupling, glycosylation) and, most of all, at present, nanocomposition [76,94,96].

## 4. Pharmacological Activity and Therapeutic Potential of Catechins—Limitations

### 4.1. Pharmacological Effects of Catechins

GTCs have been extensively studied for numerous beneficial and pharmacological effects, which can be summarized, as follows:**Direct and indirect antioxidant properties**

The antioxidant effects of GTCs have been extensively validated through in vitro, in vivo, and clinical studies [17,97,98]. The overall antioxidant capacity of green tea is further supported by additional bioactive constituents, including carotenoids, vitamins, and essential minerals, and is significantly influenced by factors such as processing methods, cultivation environment, and brewing temperature, all of which modulate catechin concentration and activity [99,100].

As mentioned above, the catechin stereoisomers contain phenolic hydroxyl groups that facilitate free radical stabilization, conferring direct antioxidant properties through hydrogen or electron transfer and subsequent formation of stabilized flavonoid radicals [101,102]. The same as the chemical stability, the antioxidant efficacy of catechins is contingent upon the number and arrangement of hydroxyl groups, which affect their interaction with reactive oxygen and nitrogen species (ROS; RNS); for example, catechin gallates display potent activity against superoxide radicals, while other flavonoids are more effective against hydroxyl radicals [103,104]. Their radical-scavenging potency typically follows the sequence EGCG > ECG > EGC > EC > C, linked to the ability to donate electrons and stabilize radical intermediates via resonance in the aromatic ring system [7,102,105]. Beyond direct scavenging, catechins also chelate transition metal ions such as Fe^2+^ and Cu^2+^, inhibiting radical generation via Fenton and Haber-Weiss reactions by binding to hydroxyl sites, particularly the catechol moiety on the B-ring [106]. This chelation effectively prevents the initiation of lipid peroxidation; however, the antioxidant behavior of flavonoids in metal-rich environments is complex and influenced by cellular context, resulting in variable activity reported in the literature [107,108]. Moreover, catechins can exert indirect antioxidant effects by enhancing the expression of endogenous antioxidant enzymes while suppressing pro-oxidant enzymes and pathways responsible for ROS production [13,107,109,110].


**Anti-inflammatory activity**


GTCs, particularly EGCG, have been extensively studied for their pronounced anti-inflammatory properties, demonstrating therapeutic potential against inflammatory diseases, such as diabetes, hypertension, chronic kidney disease, and neuroinflammation [111]. In addition to EGCG’s significant capacity to neutralize ROS and RNS (e.g., nitric oxide and peroxynitrite), a modulation of critical signaling cascades and transcription factors, including MAPK, NF-κB, AP-1, and STAT pathways, has been established [112]. Moreover, EGCG effectively suppresses inducible nitric oxide synthase and cyclooxygenase-2—key enzymes in inflammatory processes—thus reducing proinflammatory cytokine production (e.g., TNF-α, IL-1β, IL-6, IL-8); altogether, these mechanisms collectively contribute to attenuating inflammatory responses [17,113,114,115].


**Neuroprotective activity**


Due to global population aging and progressive environmental changes, neurodegenerative disorders have emerged as a major public health concern [116,117]. Alzheimer’s disease, Parkinson’s disease, Huntington’s disease, and amyotrophic lateral sclerosis represent some of the most prevalent neurodegenerative conditions, all sharing common pathogenic hallmarks—oxidative stress, mitochondrial dysfunction, genomic instability, accumulation of misfolded proteins, and chronic neuroinflammation [118,119,120]. The progression of neuroinflammation is further potentiated by factors such as aging, systemic infections, gut microbiota dysbiosis, environmental toxins, genetic mutations, and sustained activation of glial cells [117]. Neurons are particularly susceptible to oxidative damage due to their high content of polyunsaturated fatty acids, which renders them vulnerable to free radical attacks and lipid peroxidation. Additionally, elevated concentrations of iron in specific brain regions further exacerbate their sensitivity to oxidative stress. In this context, GTCs, and especially EGCG, have attracted considerable attention for their potent antioxidant, anti-inflammatory, and iron-chelating properties [120]. Beyond these well-established roles, EGCG also exerts a range of non-antioxidant neuroprotective mechanisms: modulation of calcium homeostasis; inhibition of presynaptic dopamine transporters and catechol-O-methyltransferase activity; activation of extracellular signal-regulated kinase, protein kinase C, phase II detoxifying enzymes, and endogenous antioxidant systems; as well as upregulation of genes involved in cellular survival pathways. Furthermore, EGCG influences amyloid precursor protein processing by promoting the secretion of the non-toxic sAPP-α isoform, while simultaneously inhibiting β-amyloid fibril formation and destabilizing pre-existing aggregates [119,120,121,122,123,124,125,126,127,128]. Crucially, the neuroprotective efficacy of EGCG is also supported by its capacity, albeit limited, to cross the blood–brain barrier (BBB), despite its hydrophilic nature [119,127,129]. Some authors rightly refer to this phenomenon as a “puzzle,” the unraveling of which remains an ongoing challenge on the scientific agenda [130]. A critical comparison can be drawn between the brain distribution of EGCG in its free form versus its encapsulated form. Following green tea consumption, only trace amounts of unmodified EGCG cross the BBB, where they are presumed to exert neuroprotective effects. Upon its clearance, this role may be partially assumed by EGCG metabolites, which also possess antioxidant activity [124]. The reported permeability of free EGCG across the BBB following 30 min of in vitro exposure ranges from approximately 2.8 ± 0.1% to 4.00 ± 0.17% [129,131]. Correspondingly, in vivo studies indicate that free EGCG reaches only low micromolar concentrations (up to 0.05 µM) in various brain regions following oral administration in rats [129,131,132]. Additionally, further in-depth investigations are warranted to elucidate the impact of configurational differences in the hydroxyl group at the 3-position of the flavan structure, as well as the effects of their degradation products within the brain [131,133]. Notably, physiological conditions that modulate BBB permeability, including aging, may facilitate enhanced EGCG penetration [130]. In this context, nanocarrier systems—particularly lipid-based formulations—offer distinct advantages depending on particle design, route of administration, and the specific catechin encapsulated. These systems significantly improve catechin compatibility with cellular membranes, protect against metabolic degradation and efflux-mediated elimination, and facilitate active transport via receptor-mediated pathways. As a result, nanocarriers not only increase the concentration of GTCs within brain tissue but also enable targeted and sustained release, thereby amplifying their therapeutic potential in neurodegenerative and neuroinflammatory disorders of the central nervous system (Figure 2).


**Anticarcinogenic activity**


Cancer remains a major global health challenge and the second leading cause of death among chronic non-communicable diseases [134]. Despite progress in understanding its multistep biological basis—initiation, promotion, and progression—advancing prevention, early detection, and treatment still depends on deeper insight into its underlying molecular and cellular mechanisms [135,136,137,138]. Numerous in vitro and in vivo studies, primarily in animal models, have investigated the role of GTCs in modulating carcinogenic processes, as summarized in several comprehensive reviews [100,139,140,141,142]. However, the cancer-preventive efficacy of GTCs in humans remains inconclusive. Discrepancies between in vitro and in vivo findings, along with the inherent challenges of extrapolating results from animal models to human physiology, underscore the limitations in establishing a definitive consensus on the cancer-preventive potential of green tea and other dietary phytochemicals with similar bioactivities.

The chemopreventive effects of GTCs, particularly EGCG, are mediated through multiple interrelated mechanisms targeting key hallmarks of cancer. The principal molecular and cellular pathways modulated by catechins include:

*Regulation of Cell Proliferation and Apoptosis*—GTCs modulate the cell cycle and promote programmed cell death by inhibiting cyclin-dependent kinases (CDK2, CDK4), upregulating cell cycle inhibitors such as p27^Kip1, and interacting with pro-survival proteins from the Bcl-2 family. These actions culminate in G0/G1 cell cycle arrest and the induction of apoptosis in transformed cells [142,143].

*Modulation of Oxidative Stress*—indisputably, the GTCs’ potent antioxidant activity mitigates oxidative damage to DNA, lipids, and proteins. Notably, under specific conditions, EGCG can also act as a pro-oxidant, generating intracellular ROS that contribute to apoptosis in cancer cells and activating cytoprotective pathways, such as the Nrf2/ARE signaling axis [102]. Therefore, the pro-oxidant potential of catechins may contribute to both beneficial and detrimental cellular effects. It is proposed that the mild generation of reactive oxygen species (ROS), resulting from the pro-oxidant activity of polyphenols, can stimulate the upregulation of endogenous antioxidant defenses, thereby promoting cytoprotection. Conversely, the pro-oxidant activity associated with intake of high doses of catechins may result in oxidative stress and cellular damage [102].

*Inhibition of Key Enzymes and Signaling Pathways*—EGCG interferes with multiple cancer-associated signaling cascades, including MAPK/ERK, PI3K/Akt, and AP-1 pathways. Additionally, it inhibits proteasome activity, DNA methyltransferases, and matrix metalloproteinases (MMP-2 and MMP-9), thereby suppressing tumor progression, invasion, and the epigenetic silencing of tumor suppressor genes [144,145].

*Disruption of Receptor Tyrosine Kinase Signaling*—EGCG antagonizes the activity of growth factor receptors, including EGFR, IGF1R, VEGFR2, and MET, leading to the inhibition of downstream signaling pathways responsible for cell proliferation, angiogenesis, and metastasis. In some cases, EGCG also alters membrane lipid raft dynamics, thereby affecting receptor localization and signaling [146,147].

*Direct Interaction with Molecular Targets*—Due to its polyphenolic structure, EGCG directly binds to various molecular targets, including the 67 kDa laminin receptor, vimentin, GRP78, FYN kinase, and IGF1R, as well as nucleic acids. These high-affinity interactions modulate protein function, cellular adhesion, and transcriptional regulation in tumor cells [148].

*Anti-Angiogenic Properties*—GTCs downregulate the expression of pro-angiogenic factors, such as VEGFA and FGF2, and inhibit their associated signaling pathways, resulting in impaired neovascularization and reduced nutrient supply to growing tumors [149].


**Antimicrobial activity**


GTCs have also well-documented antimicrobial activity [150,151,152]. Their antibacterial properties have been demonstrated in various in vitro studies. Catechins exhibit bactericidal activity against Gram-positive bacteria, such as *Staphylococcus aureus* and several *Staphylococcus* spp. multidrug-resistant strains, as well as Gram-negative species including *Escherichia coli*, *Pseudomonas aeruginosa*, and *Helicobacter pylori*. However, it is reported that the antibacterial activity against Gram-positive bacteria is stronger [150,153]. Moreover, synergistic effects have been reported when catechins are used in combination with conventional antimicrobial agents. The antimicrobial mechanisms of catechins appear to involve both direct and indirect actions on bacterial cells. These include modulation of gene expression, inhibition of key bacterial enzymes, as well as direct interaction with the bacterial cell membrane. Such membrane interactions can increase permeability or induce the production of hydrogen peroxide (H_2_O_2_), leading to membrane disruption and impaired bacterial adherence to host cells. Even more, GTCs have been shown to suppress the activity of bacterial toxins [153,154,155]. Specifically, in vitro studies indicate that EGCG inhibits biofilm formation and proliferation of *Streptococcus mutans*, a primary etiological agent in dental caries, potentially through effects on gene regulation and the synthesis of critical bacterial proteins [155,156]. EGCG have also demonstrated antiviral properties against a variety of DNA and RNA viruses in vitro (human herpes viruses, human papillomavirus—HPV, SARS-CoV-2). It is thought that EGCG may bind to intracellular proteins, thereby disrupting their normal functions, or may interact with cell surface receptors, consequently inhibiting viral entry. Additionally, EGCG may influence host cell gene expression [157,158,159,160]. A partially purified catechin-rich fraction derived from an aqueous extract of green tea leaves, known as sinecatechins, has been approved by the United States Food and Drug Administration for the topical treatment of external genital and perianal warts, which are primarily caused by HPV. This formulation contains 85–95% catechins by weight, with EGCG comprising more than 55% of the total catechin content. It is indicated for use in immunocompetent individuals aged 18 years and older [161,162].

### 4.2. Limitations

Despite the numerous potentially beneficial pharmacological properties of GTCs, their biological efficacy is significantly constrained by a range of factors, which may be broadly categorized into two principal groups:**Low in vitro stability**

As discussed previously, GTCs exhibit pronounced chemical instability, being susceptible to oxidative degradation, epimerization, and polymerization, particularly during the manufacturing and storage of green tea-based products [17,56,163]. Their stability is highly dependent on multiple physicochemical conditions, including initial concentration, temperature (notably unstable below 44 °C and above 98 °C), pH (marked instability at values exceeding pH 4), the presence of oxygen, and the absence of antioxidants [88,91,164,165]. Furthermore, the interaction of catechin galloyl groups with metal ions leads to the formation of complexes that diminish both their antioxidant potential and the bioavailability of essential minerals [95,166]. Industrial additives and stabilizers may further compromise catechin stability, thereby complicating their incorporation into finished products [167].


**Limited bioavailability due to pharmacokinetic constraints**


Although catechins absorption occurs predominantly in the duodenum, the alkaline environment of the small intestine and the presence of ROS favor auto-oxidative degradation [168]. Extensive metabolism begins in the small intestine, involving phase II enzymatic modifications (glucuronidation, sulfation, and methylation), which are further continued during hepatic biotransformation. A small proportion of unmetabolized catechins, together with their intestinal metabolites, may reach the colon, where they undergo further microbial degradation. Transport across the intestinal epithelium primarily occurs via passive diffusion (both paracellular and transcellular routes), in the absence of dedicated transmembrane transporters. Concurrently, active efflux mediated by multidrug resistance-associated proteins and P-glycoprotein results in the extrusion of absorbed catechins back into the intestinal lumen, thereby further reducing their systemic bioavailability [20,168,169].

Several strategies are being investigated to address these limitations, notably the encapsulation of catechins in lipid-based delivery systems to enhance their stability, absorption, and bioavailability. Such systems have shown promise in improving the in vitro and in vivo efficacy of catechins, particularly EGCG.

## 5. Perspectives Offered by Nanotechnology

For the last several decades, drug delivery has undergone a significant transformation after the introduction and development of nanotechnologies [170]. Nowadays, among the variety of submicron carriers, all designed to improve the therapeutic activity of BACs, lipid-based nanosystems stand out as multifaceted platforms. They provide the typical size-related advantages of nanoscale carriers—improved solubility, stability, and bioavailability of the active pharmaceutical ingredients (APIs) while employing reduced dosages and minimizing side effects [170,171]. These carriers, however, further exploit the biocompatibility of lipids to overcome biological barriers, additionally enhancing cellular uptake and improving the bioaccessibility of their payload [172,173]. Their tunable physicochemical characteristics also provide functional advantages in terms of relatively easy tailoring to specific targets [174,175]. Different lipid-based drug delivery systems, viz., colloidal liquid-in-liquid dispersions, vesicular carriers, and lipid nanoparticles, have been extensively used to tackle the challenges of modern therapeutics.

### 5.1. Microemulsions

Microemulsions are recognized as one of the first lipid-based nanoscale drug delivery systems. They were introduced in the 1940s by Hoar and Schulman in their study on transparent dispersions of water and oil and the conditions required for the formation thereof [176]. At their simplest, these nanoplatforms can be described as “surfactant/co-surfactant-fixed” oil and water mixtures. According to the distribution of their phases, they can be divided into four main types (Winsor classification): I—O/W (oil-in-water) microemulsion and oil phase coexisting; II—W/O (water-in-oil) system with the presence of excess aqueous phase; III—triphasic system in which bicontinuous microemulsion is in equilibrium with oleaginous and aqueous phase; IV—single-phase microemulsions, which can also be categorized as O/W, W/O, and bicontinuous [177,178] (Figure 3). Despite their resemblance to coarse emulsions in terms of qualitative composition, microemulsions exhibit different characteristics that provide significant advantages. First and foremost, these colloidal carriers are thermodynamically stable, defining their longer shelf-life [179]. Their inner phase has an average domain size of 10÷100 nm, which imparts a clear, pellucid appearance. These submicron droplets provide a large surface and contact area, thereby enhancing drug absorption and bioavailability [179,180]. The high content of emulsifiers eases the production of microemulsions—they can form readily (sometimes, even spontaneously), eliminating the need for high-energy preparation methods and facilitating their scale-up production [179,180]. The latter also affects the drug bioavailability by ensuring efficient drug solubilization and enhancing the permeability of skin and mucosal membranes [181,182]. The inclusion of high levels of surfactants and co-surfactants, unfortunately, limits the available choices and raises concerns about the toxicity and irritation potential of microemulsions [179,183]. That is why, despite their superior stability, ability to carry both hydrophilic and lipophilic APIs, and supersolubilization properties [184], the usage of these systems is often limited.

### 5.2. Nanoemulsions

Nakajima et al. initially proposed the concept of “nanoemulsion” in the early 1990s [185]. These colloidal liquid-in-liquid dispersions comprise oleaginous and aqueous phases, as well as surfactants and co-surfactants, and logically, share certain fundamental similarities with both macro- and microemulsions. Structurally and visually, nanoemulsions resemble microemulsions, as their disperse phase has submicron dimensions (typically below 200 nm), giving them a transparent appearance and ability to enhance drug absorption [186]. However, their quantitative composition is akin to that of conventional emulsions regarding the lower emulsifier/co-emulsifier concentrations [187]. Otherwise stated, the nanosystems in question can be prepared employing a wider range of surfactants, and they do not possess the toxicological drawbacks of microemulsions. Another similarity between nanoemulsions and their “coarse” counterparts lies in their classification, viz. O/W, W/O, and complex systems (O/W/O, oil-in-water-in-oil; and W/O/W, water-in-oil-in-water), with the former most commonly used [188,189] (Figure 3). In terms of stability, nanoemulsions are again comparable with macroemulsions—they are kinetically but not thermodynamically stable [190]. As a result, their preparation often requires an energy input, and they can undergo physical destabilization over time [191,192]. Although instability can occur, the risk is significantly lower in nanoemulsions compared to coarse emulsions, and their colloidal droplets can again be referred to as the “main culprits” [193].

### 5.3. Liposomes

Liposomes are among the earliest developed and most exploited lipid-based drug delivery systems. The first reports on them appear in the 1960s, in the work of Bangham et al. [194]. These vesicular nanoplatforms comprise phospholipids that undergo self-assembly into one or more spherical bilayers. Based on the number, structure, and specific arrangement of these lamellae, liposomes can be classified as uni-, oligo-, multilamellar, and multivesicular systems. The former are built up of a single phospholipid layer and can be further categorized by size into small (<100 nm), large (>100 nm), and giant (>1000 nm). Oligolamellar liposomes have an average size between 100 and 1000 nm and comprise up to five bilayer membranes, while the number of vesicles in multilamellar systems varies from five to twenty-five. Multivesicular liposomes also contain numerous lamellae, yet they are non-concentric [195,196] (Figure 3). A common characteristic of all types of liposomes is the presence of one or more enclosed aqueous chambers. Such an architecture, in which hydrophilic and lipophilic compartments are present at once, gives liposomes the advantage of encapsulating APIs that vary in nature [197]. Another favorable property of the nanosystems in question lies in their shell-like structure, which can protect the “active cargo” from potentially harmful physiological and/or environmental factors [198]. Moreover, liposomes’ mimicry with biological membranes ensures their facilitated cellular uptake, thereby enhancing drug bioavailability [199]. Their targetability can be easily achieved by surface modifications or by exploiting excipients that trigger responsiveness (i.e., drug release) to different stimuli, such as changes in pH, temperature, light, redox potential, and others [197]. Superiority can also be found in the principal composition of liposomes—since their main ingredients naturally occur in the human body, these vesicles are considered non-toxic, non-immunogenic, and biocompatible [200]. Despite their evident (and proven) potential, liposomes may suffer from several limitations—cost-ineffective manufacture and scale-up challenges, relatively low chemical stability, and the possibility of drug leakage and lamellar fusion during storage and/or after application [201].

### 5.4. Phytosomes

Another class of lipid-based carriers specifically designed to enhance the bioavailability of plant extracts and phytochemicals is that of phytosomes (also known as herbosomes). They were first developed by Bombardelli and Patri (Indena, Italy) in the late 1980s [202]. These nanosystems resemble the structure and features of liposomes, yet their bilayered membrane is constructed by phospholipid-phytoconstituent complexes (Figure 3). The latter are formed through hydrogen bonding between the polar heads of phospholipids and active H-atoms of the phytosubstrates [203]. This chemical interaction determines the distinctive structure of phytosomes and defines their beneficial properties. Unlike liposomes, where the APIs are merely entrapped within the aqueous core or into the lipid bilayer, in phytosomes, the active principles are membrane structural components. This integration significantly facilitates their absorption and bioaccessibility while reducing the risk of their leakage and improving the physical and chemical stability of the formulation [204]. However, if the interactions between the phytoconstituents and phospholipids are somehow compromised, for instance, due to unfavorable physiological conditions, the former can undergo extensive clearance [205].

### 5.5. Solid Lipid Nanoparticles

In the early 1990s, solid lipid nanoparticles (SLNs) were introduced by Lucks, Müller, and Gasco [206,207] as alternative drug carriers to the already available liposomes, microemulsions, and inorganic and polymer nanoparticles. These nanocarriers contain a surfactant-surrounded lipid core that remains solid at ambient and body temperatures [208]. Based on the drug distribution, SLNs can be divided into three structural types. Type 1 SLNs, the homogenous matrix model, possess a uniform drug distribution. Drug-enriched shell model (Type 2 SLNs) occurs when minimal drug concentrations are used, resulting in a drug-free lipid core and a drug-enriched lipid layer overlying it. When the drug reaches its maximum saturation, it concentrates in the lipid core, and SLNs Type 3 (drug-enriched core model) are formed [209] (Figure 3). These structural variations impact the drug release: SLNs of Types 1 and 3, for example, can support prolonged delivery, while Type 2 provides faster, bulk liberation [210]. Along with the potential to tailor drug release, SLNs possess other key advantages—biocompatibility, biodegradability, lack of toxicity, easy scaling, and possibility of surface modifications [211]. However, drawbacks such as low drug encapsulation capacity and rapid expulsion, related to the solid lipids’ high degree of crystallinity and possible lipid polymorphic transitions, are inherent to the nanosystems in question [212]. Despite these challenges, SLNs remain promising nanocarriers with potential in targeted and controlled drug delivery, evidenced by the ongoing research for their implementation in cancer therapy, neurodegenerative diseases, gene delivery, and treatment of various infectious diseases [213,214,215,216].

### 5.6. Nanostructured Lipid Carriers

Nanostructured lipid carriers (NLCs), often referred to as “second-generation SLNs”, first appeared around the 2000s in the work of Müller et al. [217]. In contrast to their predecessors, NLCs possess increased physical stability, higher drug-entrapping capacity, and minimized drug expulsion during storage. These advantages stem from the liquid oils included in their lipid cores [208,218]. Depending on the similarity between the employed oleaginous components, as well as the ratios thereof, three types of NLCs can be distinguished. Imperfect NLCs (Type 1) are obtained by mixing solid lipids with small amounts of incompatible liquid oils. As a result, an imperfect crystalline matrix with multiple cavities for accommodating APIs is arranged. Type 2 (amorphous NLCs) comprises solid lipids that do not recrystallize after cooling. Type 3, or multiple type NLCs, are produced by incorporating liquid oils in concentrations exceeding their solubility in the solid lipid, forming numerous drug-containing oil nanodroplets in the solid scaffold and providing a prolonged drug release [211,219] (Figure 3). Alas, NLCs do not come without demerits. They can still encounter physical instability in various forms, such as particle aggregation, creaming, or gelling over time [220]. Moreover, the complexity of their lipid matrices hinders their batch-to-batch reproducibility and large-scale manufacturing [221]. Nonetheless, NLCs offer enough valuable benefits to continue to be widely exploited, especially in the delivery of poorly water-soluble drugs, paving the way for new therapeutic approaches.

### 5.7. Other Lipid-Based Drug Delivery Systems

Apart from the most recognizable and utilized lipid-based nanosystems described so far, several other nanoplatforms have also taken their place in the field of drug delivery.

Transferosomes, pliant liposome derivatives introduced at the end of the 20th century by Cevc [222], are characterized with significant potential as transdermal delivery systems. Except for phospholipids, these vesicular carriers contain the so-called edge activators—single-chain surfactants that “cause” ultra-deformability of the bilayer membrane. The resultant flexible lamellae provide increased permeability while minimizing the possibility of rupture and drug leakage and ensuring the higher physical stability of transferosomes [223,224].

Niosomes, another type of vesicle carriers structurally resembling liposomes, utilize non-ionic surfactants and lipidic molecules (e.g., cholesterol) in their bilayered membranes. These self-assembling structures were patented in the 1970s by L’Oréal, France [225]. Except for the common advantages they share with liposomes (biocompatibility, biodegradability, lack of toxicity and immunogenicity, and ability to encapsulate both lipophilic and hydrophilic BACs), niosomes offer higher chemical and physical stability, as well as cost-effective production methods. That is why their employment has expanded far beyond the cosmetic industry throughout the years, and nowadays, they are extensively used as drug delivery systems for a myriad of APIs through various routes of administration [226,227].

Hexosomes are non-lamellar lipid nanoparticles consisting of hexagonally arranged inverse micelles. The latter are formed spontaneously by dispersing polar amphiphilic lipids with limited aqueous solubility in water. These nanosystems are also capable of delivering BACs of different natures, improving their solubility, and enhancing their permeation [228,229].

All advantages provided by lipid-based nanocarriers are highly valuable regarding the varying physicochemical properties of major catechin representatives, their limited chemical stability, poor permeability, and generally low bioavailability. Currently, no GTC lipid nanocarriers are approved for clinical use. Despite promising results from preclinical studies, these formulations have yet to advance to clinical trials in humans. To gain regulatory approval, developers must comply with established requirements for lipid-based therapeutic products, which include detailed evaluation of chemistry, manufacturing, and control, comprehensive safety assessment, and thorough clinical validation [230,231,232]. However, all of the described nanosystems have been making inroads in the delivery of catechins, highlighting their pharmacotherapeutic potential.

## 6. Lipid-Based Nanotechnologies for Drug Delivery of GTCs in the Search of Improved Anti-Inflammatory and Antioxidant Activity

Zhang et al. developed EGCG-loaded NLCs and chitosan-coated NLCs designed to enhance EGCG’s stability, bioavailability, and bioactivity in macrophages associated with atherosclerosis. These nanocarriers were prepared using natural triglycerides, soy lecithin, and the surfactant Kolliphor HS15, with chitosan added to improve cellular uptake. In vitro assays demonstrated that chitosan-coated NLCs significantly increased EGCG content in THP-1-derived macrophages compared to free EGCG, and improved EGCG stability across various pH levels and temperatures. Nanoencapsulation also enabled sustained release, with less than 5% EGCG released from NLCs over a 9 h period, whereas non-encapsulated EGCG underwent rapid degradation. Functionally, both NLCs and chitosan-coated NLCs effectively reduced cholesteryl ester accumulation in macrophages and suppressed expression of proinflammatory markers. Specifically, NLCs decreased MCP-1 mRNA levels, and chitosan-coated NLCs significantly reduced MCP-1 secretion—an inflammatory chemokine critical for monocyte recruitment and atherosclerotic plaque development. These effects were not observed for TNF-α or IL-6 levels. Given that elevated MCP-1 is a validated biomarker of vascular inflammation and atherosclerosis in humans, these findings highlight the potential of EGCG-loaded nanocarriers, as promising anti-inflammatory and anti-atherogenic agents targeting macrophage-mediated pathways [233].

Further enhancing targeted delivery, the same research group of Zhang et al. innovatively introduced CD36-targeted LNs, encapsulating EGCG to enhance targeted delivery to intimal macrophages implicated in atherosclerosis. The nanoparticles were engineered by integrating KOdiA-PC, a specific CD36 ligand, resulting in improved macrophage-specific uptake. These nanocarriers were designed by functionalizing EGCG-loaded vesicles with 1-(Palmitoyl)-2-(5-keto-6-octene-dioyl) phosphatidylcholine (KOdiA-PC), a high-affinity CD36 ligand, to facilitate targeted uptake by intimal macrophages. In vitro assays confirmed that the elaborated LNs selectively interacted with macrophages via the CD36 receptor, as validated through antibody blocking and receptor knockdown experiments. In LDL receptor-null (LDLr^−/−^) mice, treatment with the nanocarriers significantly reduced levels of proinflammatory cytokines, including monocyte chemoattractant protein-1, TNF-α, and IL-6, as well as the aortic lesion area, outperforming both free EGCG and non-targeted nanoparticle controls. Additionally, CD36-targeted LNs exhibited improved EGCG stability and targeted delivery, with reduced liver accumulation relative to native EGCG, positioning it as a promising strategy for macrophage-targeted therapy in atherosclerosis and other macrophage-driven inflammatory conditions [234].

In ocular applications, Fangueiro et al. successfully synthesized biocompatible and biodegradable cationic lipid nanoparticles incorporating EGCG, to overcome ocular drug delivery limitations such as rapid clearance and poor permeability and low bioavailability. These nanoparticles were formulated using natural and physiological lipids, cationic surfactants such as cetyltrimethylammonium bromide (CTAB) or dimethyldioctadecylammonium bromide (DDAB), and water, to ensure controlled and prolonged release of EGCG while maintaining therapeutic concentrations over extended periods. Ex vivo permeation studies through corneal and scleral tissues demonstrated effective transcorneal and transscleral delivery, suggesting potential for EGCG absorption in both anterior and posterior ocular segments. Corneal permeation followed first-order kinetics in both tested formulations, while CTAB-based LNs followed a Boltzmann sigmoidal profile and DDAB-containing LNs a first order kinetics profile. Biocompatibility was confirmed through the in vitro HET-CAM assay and in vivo Draize test, both indicating a safe and non-irritant profile. The cationic nature of the nanoparticles likely contributed to higher drug residence time, higher drug absorption and consequently higher bioavailability of EGCG in the ocular mucosa. These findings underscore the therapeutic potential of EGCG-loaded lipid nanoparticles in the management of oxidative stress- and inflammation-driven ocular pathologies such as diabetic retinopathy, age-related macular degeneration, and macular edema [235].

It is pertinent to note that cationic surfactants such as CTAB and DDAB differ significantly in their safety profiles. CTAB exhibits pronounced cytotoxicity even at low concentrations, primarily through disruption of cellular membranes and induction of oxidative stress and apoptosis. In contrast, DDAB demonstrates markedly greater biocompatibility, maintaining high cell viability at substantially higher doses, attributable to its more benign two-tailed lipid structure [236,237,238,239]. Nevertheless, cationic nanoparticles inherently pose risks of membrane damage, inflammatory responses, and immune activation [240,241]. Additionally, solid lipid nanoparticles have been associated with oxidative stress and altered antioxidant enzyme activity in preclinical models [242,243]. Consequently, thorough toxicological evaluation—including assessments of cytotoxicity, oxidative stress, and immunological effects—is essential for the development of safe GTC nanoformulations.

For topical dermal application, Harwansh et al. designed a catechin-loaded nanoemulsions-based nano-gel to enhance the antioxidant and photoprotective effects of C against UVA-induced oxidative stress. The nanoemulsion was prepared using ethyl oleate as the oil phase, and a surfactant/co-surfactant system. The nano-gel demonstrated significantly improved skin permeability (96.62%) compared to conventional C gel (53.01%) over 24 h and exhibited significantly enhanced relative bioavailability (894.73%) after transdermal application. In vivo studies in rats revealed that nano-gel markedly restored the levels of endogenous antioxidant enzymes like superoxide dismutase (SOD), glutathione peroxidase (GSH-Px), and catalase, and reduced thiobarbituric acid reactive substances in skin tissues exposed to UVA radiation, outperforming the conventional gel formulation. The nano-gel also exhibited prolonged C release, enhanced stability, and no skin irritation, supporting its potential as a topical nanocarrier system for antioxidant and anti-inflammatory photoprotection [244].

Addressing oral administration challenges, Liang et al. formulated EGCG-loaded niosomes, composed of Tween 60 and cholesterol, to improve catechin stability and antioxidant activity in gastrointestinal conditions. These nanocarriers, prepared via the ethanol injection method, achieved an encapsulation efficiency of ~76% and exhibited a uniform spherical morphology with an average diameter of ~60 nm. In simulated gastrointestinal conditions, niosomal encapsulation significantly enhanced EGCG stability: residual EGCG after 2 h in simulated intestinal fluid increased from 3% (free EGCG) to 49% (niosomal EGCG). The protective effect was attributed to the resistance of niosomes to enzymatic degradation, especially against pancreatin. In vitro antioxidant assays demonstrated that EGCG-loaded niosomes had greater ferric reducing antioxidant power and higher cellular antioxidant activity in human hepatocellular carcinoma HepG2 cells compared to free EGCG, both before and after digestion. These findings support niosomes as a promising delivery system to enhance the oral bioavailability and bioefficacy of catechins like EGCG, especially by protecting against degradation and improving cellular antioxidant response [245].

Furthering these findings, Shariare et al. introduced a phytosomal EGCG formulation comprising egg phospholipids and cholesterol, developed via an optimized solvent injection method and optimized via design of experiments. This formulation exhibited high drug loading (up to 90%) and demonstrated significant anti-inflammatory effects, exhibiting up to 88.2% inhibition in carrageenan-induced paw edema in rats, thus notably surpassing both green tea extract and free EGCG. The enhanced bioavailability and sustained therapeutic action of these nanophytosomes highlight the clinical potential of lipid-based nanodelivery systems in chronic inflammatory therapies [246].

These studies collectively highlight the transformative impact of nanotechnology in unlocking the full therapeutic potential of catechins. By enhancing bioavailability, optimizing tissue targeting, and prolonging anti-inflammatory effects, nanoformulated catechins represent a promising approach for the treatment of both systemic and localized inflammatory conditions. Table 2 provides a summary of the above-mentioned and additional research efforts focusing on lipid-based nanocarriers for catechin delivery, highlighting their antioxidant and anti-inflammatory potential in both in vitro and in vivo models [247,248,249,250].

The dual antioxidant/pro-oxidant activity of GTC has been previously discussed in this work. In this context, lipid-based nanocarriers modulate this balance primarily by regulating local concentration, stability, and cellular delivery of catechins. Encapsulation within lipid carriers protects catechins from premature oxidation and degradation in physiological environments, thereby preserving their antioxidant potential. Conversely, controlled release from the lipid matrix can prevent excessive local concentrations that might otherwise trigger pro-oxidant effects through redox cycling and ROS generation. Moreover, lipid nanocarriers can enhance intracellular delivery, potentially promoting targeted pro-oxidant activity in diseased cells, such as cancer cells, where elevated ROS levels may induce apoptosis. Thus, lipid carriers can be regarded not only as delivery vehicles but also as modulators capable of both mitigating unwanted systemic pro-oxidant effects and enabling localized pro-oxidant mechanisms beneficial for therapeutic outcomes.

## 7. Lipid-Based Nanotechnologies for Drug Delivery of GTCs in the Search for Improved Neuroprotective Activity

Rivera et al. utilized liposomes as a strategy to enhance the brain delivery of antioxidants, including C (30 mg/kg), and to subsequently investigate their neuroprotective effects in an in vivo model of brain damage induced by focal ischemia in rats. Chromatographic analysis revealed that the amount of C reaching brain tissue following a single i.p. administration of liposomal preparations was 10.5 ng/g, while in aqueous preparations, it was below the limit of detection. On the other hand, in the in vivo study, only the C-loaded liposomes failed to provide protection with respect to brain tissue [251]. However, it would be of interest to explore the neuroprotective effects at higher C doses, as well as with liposomes loaded with other catechins.

Huang et al. also employed liposomes for the encapsulation of C, aiming to enhance its bioavailability and target its delivery to the brain. The prepared carriers were elastic liposomes (mean size 35 ÷ 70 nm), fabricated using the thin-film method followed by ultrasonic treatment and extrusion. These liposomes contained soybean phosphatidylcholine, cholesterol, and Tween 80 in the presence of 15% ethanol. In vitro experiments revealed that liposomes containing C exhibited a prolonged release profile, while maintaining their stability under simulated intestinal fluid conditions (compared to an aqueous solution). The promising results were confirmed in vivo following oral treatment in rats. Blood levels of liposomal C were elevated at a later stage post-administration, compared to the free form. Specifically, it showed 2.9- and 2.7-fold higher accumulation of the antioxidant in the cerebral cortex and hippocampus, respectively, compared to the aqueous solution. Increased concentrations of the compound were also observed in the striatum and thalamus, suggesting that such nanodelivery systems may be effectively explored in studies concerning neurodegenerative diseases [252].

Cheng et al. presented a modified liposomal formulation encapsulating EGCG. They focused their study on the inhibition of microglia-mediated inflammation, given its role in the progression of neurodegenerative diseases. The particles were prepared from phosphatidylcholine or phosphatidylserine, with or without a vitamin E coating, through a hydration method followed by extrusion through a membrane. The authors report that liposomes containing phosphatidylserine were smaller, more stable, exhibited higher encapsulation efficiency, and that the addition of vitamin E further protected EGCG from oxidation, thereby enhancing the encapsulation efficiency. Using a cellular model of lipopolysaccharide-induced inflammation in BV-2 microglial cells, it was found that the expression of TNF-α and the production of nitric oxide were reduced following pre-treatment with catechin-loaded liposomes, indicating inhibition of the neuroinflammatory response. The results provided a rationale for the authors to validate their findings under in vivo conditions. In addition, a rat model of Parkinson’s disease was employed, in which the condition was induced by unilateral injection of LPS into the substantia nigra of the midbrain. Post-treatment with EGCG-loaded liposomes led to symptomatic improvement, suppression of neuroinflammation, and a reduction in TNF-α secretion [253].

To overcome the limitations of liposomes, such as sedimentation, aggregation, and oxidation, Al-Najjar et al. propose EGCG-loaded proliposomal vesicles [254]. These phospholipid-based systems enhance oral bioavailability by protecting compounds from premature gastrointestinal degradation and mimicking biological membranes [255,256]. In a rat model of traumatic brain injury, seven-day pre-treatment with EGCG and EGCG-proliposomes (equivalent doses) led to a significant reduction in the lipid peroxidation marker malondialdehyde (MDA) (*p* < 0.05), with EGCG-proliposomes showing a stronger effect. Brain tissue from treated animals showed a significant increase in antioxidants glutathione and superoxide dismutase (*p* < 0.05), with EGCG-proliposomes again showing better results. Additionally, activation of the Sirt1/Nrf2/HO-1 pathway was more pronounced compared to free EGCG. Immunohistochemical analysis revealed higher HO-1 protein expression (*p* < 0.05) in the cerebral cortex and hippocampus, further validated by histopathological analysis, confirming the neuroprotective role of lipid-based nanoparticles [254].

Another study reaching the in vitro stage investigated the cytotoxicity, brain-targeting capability, and antioxidant neuroprotective effects of glucose-modified liposomes encapsulating EGCG using bEnd.3 and PC12 cell lines. By incorporating a glucose ligand and optimizing the ratios of EGCG, lipids, soybean phospholipids, and cholesterol, the research group of Xia et al. obtained a glucose-modified liposomal formulation with high encapsulation efficiency, an optimal particle size (average diameter of 158.7 nm) for brain targeting, and satisfactory stability. The formulation demonstrated reduced cytotoxicity and enhanced protection against H_2_O_2_-induced oxidative stress compared to free EGCG and unmodified liposomes, while maintaining ROS levels comparable to the control. The increased cellular uptake and improved permeability across the BBB via GLUT1 transporters underscore the potential of this carrier for efficient delivery to brain tissue, positioning it as a promising candidate for further investigation in pharmaceutical or functional food applications [257].

In another recent in vitro study of Kuo et al., the additive properties of resveratrol and EGCG encapsulated in complex liposomes were investigated. The particles were assembled using 1,2-distearoyl-sn-glycero-3-phosphocholine, dihexadecylphosphate, cholesterol, and 1-palmitoyl-2-oleoyl-sn-glycero-3-phosphate. The surface of the carriers was modified with leptin to facilitate passage through the BBB, with the aim of restoring degenerating dopaminergic neurons. Immunofluorescence analysis revealed that this modification enabled the liposomes to bind to HBMECs and SH-SY5Y cells via the leptin receptor, enhancing their ability to cross the BBB and be absorbed by the cells. Additionally, reductions in the apoptosis-promoting protein Bcl-2-associated X protein and α-synuclein were observed, along with an increase in the apoptosis-inhibitory protein B-cell lymphoma 2, tyrosine hydroxylase, and the dopamine transporter, highlighting the need to test the anti-Parkinsonian effects of the liposomes in in vivo models [258].

Other variants of lipid-based nanodelivery systems have also been tested to investigate and enhance the neuroprotective properties of catechins. Smith et al. investigated the potential of nanolipidic particle complexes to enhance the therapeutic efficacy of EGCG in the treatment of Alzheimer’s Disease and/or HIV-associated dementia. Their findings demonstrated that the formation of nanolipidic EGCG particles augmented the neuronal α-secretase activity in vitro by up to 91%. In a subsequent in vivo study conducted on male Sprague-Dawley rats, which were orally administered the same EGCG particles (containing 100 mg EGCG/kg body weight), bioavailability was observed to be more than two-fold greater compared to free EGCG [259].

Mishra et al. proposed an efficient strategy to enhance the bioavailability and targeted delivery of EGCG to the brain using transferosomes (corresponding to 0.5 mg/kg body weight of EGCG), prepared via the thin-film hydration method. These specialized vesicles consist of concentric layers of phosphatidylcholine—a principal phospholipid in eukaryotic membranes, cholesterol, and an edge activator, surrounding an aqueous or ethanolic core. Designed to potentiate the neuroprotective effects of EGCG, the system also facilitates assessment of its synergistic potential with ascorbic acid, a compound of promising relevance to Alzheimer’s disease therapy. Following intranasal administration in mice, pharmacokinetic comparisons revealed that the transferosome-loaded formulations achieved approximately a fivefold increase in long-term brain concentrations of EGCG compared to its free form. In a mouse model of Alzheimer’s disease, intranasal treatment with EGCG-ascorbic acid-transferosomes resulted in enhanced acetylcholinesterase activity, reduced neuroinflammation, and improvements in spatial learning and memory. These findings underscore the therapeutic potential of this novel formulation in the treatment of neurodegenerative disorders [260].

Using a microemulsification technique, Kaur et al. developed EGCG-loaded SLNs with a mean diameter of 162.4 nm and a spherical morphology. The in vitro release profile of the catechin was sustained and gradual. Enhanced brain bioavailability of EGCG, as demonstrated in the study, was associated with significant improvement in memory impairments in a mouse model of cerebral ischemia [261]. Likewise, Nunes developed SLNs as a delivery system for EGCG and/or vanillic acid. The study demonstrated technological success, while the authors identified in vivo stability of the natural compounds and investigation of their potential synergistic effects in the prevention or treatment of Alzheimer’s disease as important future directions—advancements that would be of considerable scientific interest. Numerous other studies have also reported promising outcomes, either in terms of the physicochemical stability or regarding the pharmacokinetic properties and safety profile of catechins encapsulated in lipid-based nanocarriers [169,246,262,263,264,265,266,267,268,269,270,271]. Nevertheless, further in-depth investigation is required to determine their potential as neuroprotective agents in vivo, as well as under the conditions of clinical studies in humans. Table 3 summarizes the characteristics of various lipid-based nanocarriers for delivering green tea catechins with demonstrated neuroprotective effects.

Despite the promising results, a crucial issue regarding the doses required to achieve neuroprotective effects and the safety profile of green tea catechins has arisen. Using linear approximation, Smith et al. conclude that an oral dose of 1800 mg/70 kg/day EGCG would be necessary to achieve therapeutically effective plasma concentrations of EGCG that would yield health benefits for patients with relevant neurodegenerative indications [259]. On the other hand, the European Food Safety Authority warns that daily EGCG intake should not exceed 800 mg, as higher doses have been associated with increased serum transaminase levels, indicating liver damage [272]. Norway also reviewed potential safety concerns related to green tea extract consumption and concluded that a daily intake exceeding 0.4 mg EGCG/kg body weight as a bolus could cause adverse biological effects. Furthermore, it was noted that there is an increased susceptibility to toxicity when green tea extract is consumed following fasting [273]. Therefore, the optimal dosing of EGCG in humans remains an important issue in the field of clinical trials, regardless of whether the substance is in its pure form or encapsulated in nanocarriers (including lipid-based ones) [17,102,120]. The analysis of existing studies further underscores the necessity for long-term safety assessments of catechin-based formulations, particularly in the context of chronic therapeutic regimens. In this context, the encapsulation of catechins within nanocarrier drug delivery systems has emerged as a key strategy for achieving enhanced therapeutic efficacy at lower dosing levels [274].

## 8. Lipid-Based Nanotechnologies for Drug Delivery of GTCs in the Search of Improved Anticarcinogenic Activity

Ramesh and Mandal reported a notable increase in the bioavailability of EGCG-loaded SLNs, which effectively protected EGCG from degradation [267]. The formulation exhibited a particle size of approximately 300 nm, encapsulation efficiency of 81%, and a sustained release profile following Higuchi kinetics via Fickian diffusion. Pharmacokinetic studies in rats demonstrated significantly higher plasma concentrations and a larger area under the curve for EGCG-loaded SLNs compared to free EGCG, indicating improved systemic exposure. Tissue distribution studies confirmed enhanced EGCG accumulation in various organs, and toxicokinetic analyses revealed no adverse effects in both acute and sub-chronic settings. These findings support EGCG-comprising SLNs as a safe and effective nanocarrier for oral delivery of EGCG. Further advancing this approach, Radhakrishnan et al. developed SLNs conjugated with the bombesin peptide to target gastrin-releasing peptide receptors, which are overexpressed in breast cancer cells [275]. The bombesin-conjugated SLNs enhanced EGCG stability and cellular uptake via receptor-mediated endocytosis, leading to significantly lower IC_50_ values and higher apoptosis rates in vitro compared to free EGCG or unconjugated SLNs. In vivo, treatment with the nanocarriers in a murine melanoma model resulted in reduced tumor volume and prolonged survival, demonstrating the potential of this targeted delivery strategy.

In another study, Silva et al. evaluated EGCG-loaded cationic SLNs against MCF-7 human breast cancer cells. These nanoplatforms significantly reduced cell viability in a dose-dependent manner, with an IC_50_ value lower than that of free EGCG. This enhanced cytotoxicity was attributed to improved cellular uptake and sustained drug release, enhancing the overall bioactivity of EGCG [276].

de Pace et al. demonstrated that chitosan-coated liposomes markedly improved EGCG stability, intracellular accumulation, and anticancer efficacy in MCF7 cells [277]. They retained antiproliferative and proapoptotic activity even at concentrations as low as 10 µM, a level at which native EGCG is ineffective. Additionally, nanoencapsulation protected against degradation and facilitated sustained release, resulting in over a 30-fold increase in intracellular EGCG levels. Collectively, these and other studies presented in Table 4 underscore the favorable pharmacokinetic and pharmacodynamic profiles of lipid-based delivery systems in enhancing the anticarcinogenic potential of catechins. Such systems represent a promising platform for translating catechin-based chemoprevention into clinical applications.

Table 4 summarizes the characteristics of various lipid-based nanocarriers for delivering green tea catechins with demonstrated anticarcinogenic activity in in vitro and/or in vivo studies.

## 9. Lipid-Based Nanotechnologies for Drug Delivery of GTCs in the Search of Improved Antimicrobial Activity

In the scientific literature are reported different methods for increasing GTCs bioavailability and stability such as incorporation in aqueous nanoparticles as well as lipid-based drug delivery systems [284,285]. However, experimental data about the antimicrobial activity of lipid-based formulations containing catechins is limited (Table 5).

Gharib et al. prepared EGCG-loaded liposomes with neutral, positive and negative surface charges and evaluated their efficacy against methicillin-resistant *S. aureus* (MRSA) in vitro and in vivo. The liposomes were prepared from an egg lecithin and cholesterol (5:1) by an extrusion method. The minimum inhibitory concentration (MIC) of the cationic EGCG-loaded nanoliposomes against MRSA was 16 mg/L compared to 256 mg/L and 128 mg/L for anionic nanoliposomes and free EGCG, respectively. The cationic liposomes also showed higher antimicrobial activity in mice infected by MRSA inoculum injected subcutaneously into a burned skin [286]. The enhanced antimicrobial efficacy of cationic EGCG-loaded nanoliposomes is primarily due to their strong electrostatic affinity for bacterial membranes, which facilitates more efficient EGCG delivery, disrupts membrane integrity, and ensures higher local drug concentrations—advantages that are not as pronounced with anionic carriers or free EGCG.

The study of Moreno et al. demonstrated that EGCG-loaded lipid-chitosan hybrid nanoparticles possess antibacterial activity against planktonic microorganisms in vitro. The lipid-chitosan hybrid nanoparticles were prepared from poloxamer (Pluronic^®^ F68), chitosan, and illipe butter by an emulsification and sonication method. The established MIC of the EGCG-loaded nanoparticles was 33.75 µg/mL for *Streptococcus mutans* and *Streptococcus sobrinus*, and 67.5 µg/mL for *Lactobacillus casei* [287].

Zou et al. prepared liposomes containing GTCs by an ethanol injection−dynamic high-pressure microfluidization method. The tea polyphenol mixture was composed of EGCG (50.44%), ECG (13.84%), EGC (13.76%), EC (4.98%), C (4.56%), gallocatechin gallate (2.02%), other flavonoids, anthocyanins and phenolic acids. Phospholipid, cholesterol, Tween 80, and GTCs were used for the preparation of nanoliposomes. The GTCs encapsulated in nanoliposomes showed smaller inhibition zones (mm), as well as higher MIC for *Salmonella typhimurium*, *Listeria monocytogenes*, *Staphylococcus aureus*, and *Escherichia coli* compared to tea GTCs in solution. According to the authors, this result is linked to the characteristics of the prepared liposomes [288].

SLNs containing green tea extract composed of polyphenols, GTCs, and caffeine were prepared by Manea et al. using high shear homogenization method. Nanoparticles were obtained using cetyl palmitate, glyceryl stearate, Tween 20, Tween 80, and lecithin. The green tea extract-loaded SLNs showed antioxidant and antibacterial activity against *Escherichia coli* [289].

Sinsinwar and Vadivel prepared C-in-cyclodextrin-in-phospholipid liposomes. C was first encapsulated with β-cyclodextrin and subsequently by soybean lecithin. The prepared liposomes exhibited in vitro and in vivo antibacterial activity against MRSA [290].

## 10. Conclusions

GTCs are among the most explored and investigated phytoconstituents because there are, figuratively speaking, countless reports on their explicit antioxidant, anticancer, neuroprotective, anti-inflammatory, antimicrobial, and else pharmacological properties. The current research and development is mostly directed at ensuring a desirable level of chemical stability, absorption, and thereby enhanced bioavailability of these BACs so that they can exert their beneficial effects in vivo. The lipid-based nanotechnologies are key to the successful drug delivery for many APIs, and as the findings of this review testify—the catechins are no exception. Amidst the numerous nanotechnological approaches, the ones comprising natural or synthetic lipids appear as the safest and enhance the permeation of therapeutic agents with poor penetrability the most; therefore, we believe, they deserve particular attention when it comes to novel delivery systems of catechins. Although numerous lipid-based nanosystems have demonstrated significant enhancements in bioavailability in animal studies, none of the lipid-encapsulated GTC formulations have advanced to clinical evaluation in humans. The encouraging preclinical findings highlight considerable therapeutic potential, yet their validation in human trials remains a task for future research. Numerous studies included in our research prove the benefits of using lipid-based nanocarriers as delivery platforms for GTCs. The summarized scientific reports emphasize chemical stabilization, improved cellular uptake, successful passive or active targeting, and overall, fortified pharmacological activity, potentially valuable in many therapeutic directions. Still, the nature of the lipid-based vesicular and particulate systems sets challenges to physicochemical stability, reproducibility, and cost-effectiveness. It is not to be neglected the possibility of a shielding effect of the carrier on the activity. The development of lipid-based carriers for GTCs appears to be a very precise and demanding process in which many requirements should be met. These include mild preparation conditions for chemical preservation of the compounds; a short, minimalistic, and effective process of drug loading for the establishment of reproducible technology; sufficient drug release for the manifestation of the pharmacological activity; and sustainability and biocompatibility. It is clear that the evolution of lipid-based nanotechnologies manages to make progress in all aspects when it comes to GTCs but still, the need exists for further research and strict control over the observed effects and phenomena. Based on the analysis of the consolidated data, it can be reasonably concluded that in vitro studies, while useful, rely on simplified models that do not capture the complex biological interactions of living organisms, such as immune responses, metabolism, and systemic distribution. Consequently, they often overestimate the efficacy and stability of nanocarriers. Although in vitro results show improved cellular uptake and controlled release of GTCs, in vivo bioavailability is limited by absorption barriers, enzymatic degradation, and clearance mechanisms. In vitro cytotoxicity does not always predict in vivo toxicity, which is influenced by organ-specific accumulation and immune responses. Release kinetics, stable in vitro, may be disrupted in vivo by factors like pH changes and enzyme activity. The antioxidant vs. pro-oxidant balance of GTCs also differs between settings due to local concentrations and oxidative stress. Moreover, formulations stable in vitro may face issues with scale-up and storage in physiological conditions. Therefore, in vitro and ex vivo findings must be cautiously interpreted and validated by well-designed in vivo studies to ensure accurate assessment of nanocarrier performance and support clinical translation.

## Figures and Tables

**Figure 1 pharmaceutics-17-00985-f001:**
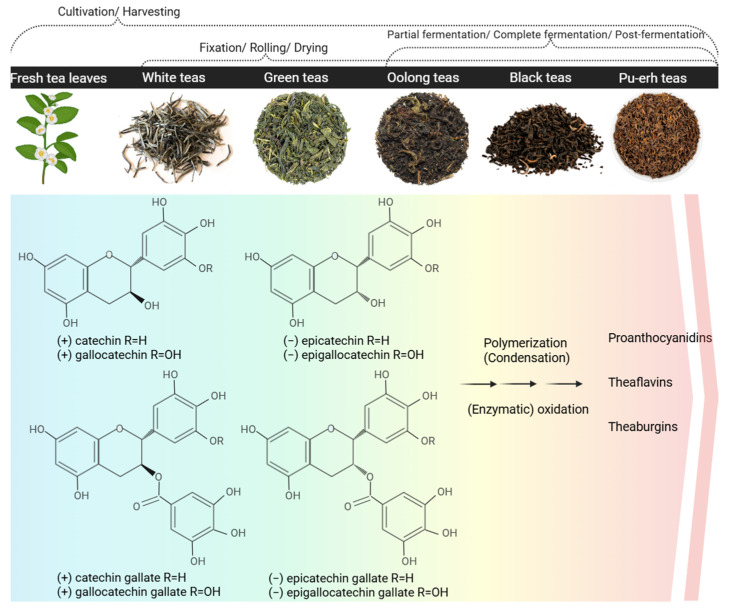
Chemical structures of catechins and *C. sinensis* teas as the primary sources for their isolation. Created in BioRender. Ivanova, N. (2025) https://BioRender.com/ugrvxl0.

**Figure 2 pharmaceutics-17-00985-f002:**
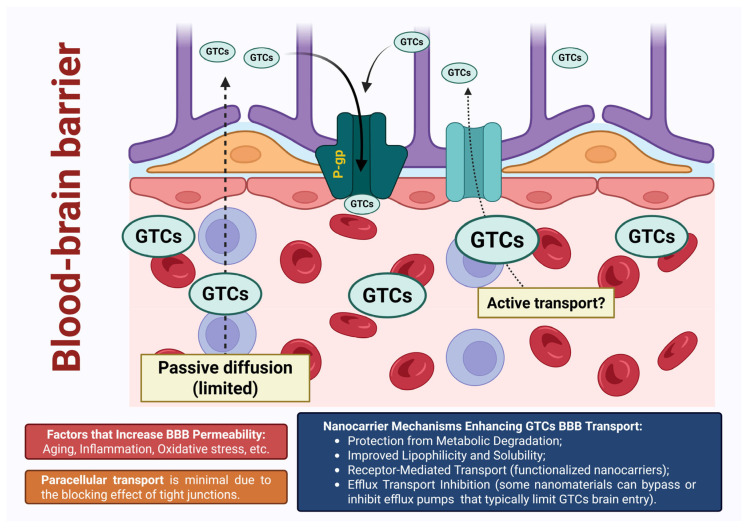
Schematic representation of the mechanisms by which GTCs cross the BBB: While free GTCs exhibit limited passive diffusion and are subject to efflux, nanoformulated catechins demonstrate enhanced brain penetration. Created in BioRender. Stoeva-Grigorova, S. (2025) https://BioRender.com/0qultjs.

**Figure 3 pharmaceutics-17-00985-f003:**
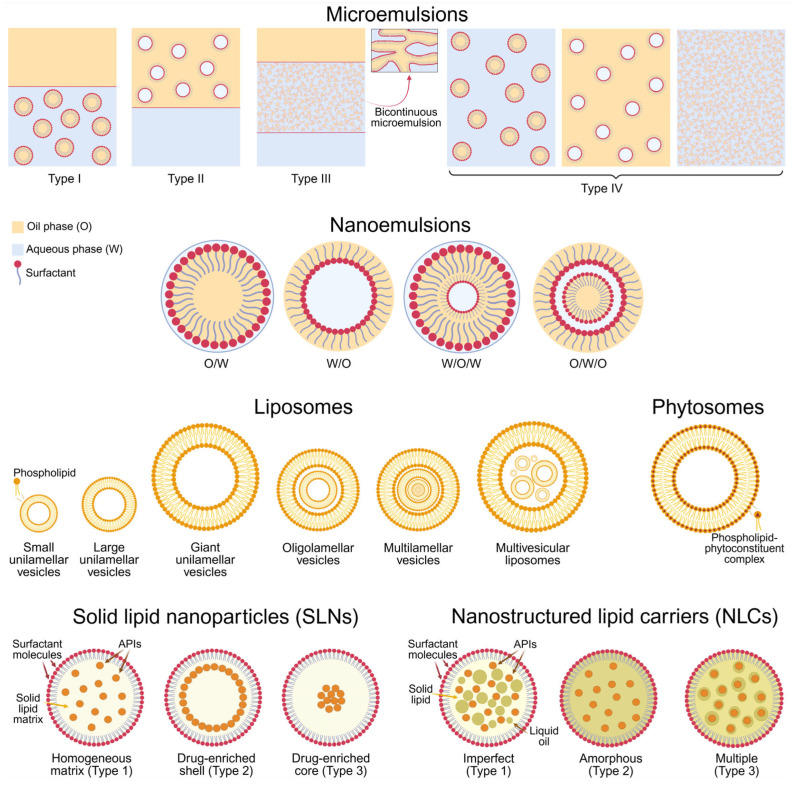
Schematic representation of different lipid-based drug delivery nanosystems with their subtypes. Created in BioRender. Ivanova, N. (2025) https://BioRender.com/pkkk0zv.

**Table 1 pharmaceutics-17-00985-t001:** Physicochemical parameters of the most common GTCs [76].

GTC	Hydroxyl Groups Count	H-Bond Donor Count	H-Bond Acceptor Count	Molecular Weight (g/mol)	Solubility Data
C	5	5	6	290	In water at 25.6 °C: 2.26 g/L [78] In water: approx. 1.8 g/L ^ii^ [79] In PBS ^i^ 7.2: approx. 1.6 g/L ^ii^ [80] In ethanol: approx. 100 g/L ^ii^ [80]
EC	5	5	6	290	Data not found
EGC	6	6	7	306	Data not found
ECG	7	7	10	442	Data not found
EGCG	8	8	11	458	In water at 20 °C: 40 g/L [81]

^i^ PBS—Phosphate-buffer saline; ^ii^ temperature not specified.

**Table 2 pharmaceutics-17-00985-t002:** Characteristics of various lipid-based nanocarriers for delivering green tea catechins with demonstrated anti-inflammatory and antioxidant effects in in vitro and/or in vivo studies.

Active Compound/s	Nano-Carrier Type	Nano-Carrier Characteristics	Suggested Mechanism/s of Action	Scientific Result	Reference
EGCG	NLCs, with or without chitosan coating	Mean size: NLCs: 46.3 nm Chitosan-coated NLCs: 53.5 nm Polydispersity index: NLCs: 0.19 Chitosan-coated NLCs: 0.19 ζ-potential: NLCs: −12.6 mV Chitosan-coated NLCs: +13.3 mV Entrapment efficiency (NLCs and chitosan-coated NLCs): ~99% Loading capacity (NLCs and chitosan-coated NLCs): ~3%	Both EGCG-loaded NLCs and chitosan-coated NLCs effectively reduced cholesteryl ester accumulation and downregulated MCP-1 expression, suggesting their strong potential to mitigate inflammatory responses and slow or even reverse the progression of atherosclerotic lesions.	Encapsulation of EGCG in NLCs and chitosan-coated NLCs improved EGCG stability; Provided sustained release of EGCG; Increased intracellular concentrations of EGCG in THP-1-derived macrophages: - Free EGCG: 0.098 µg/mg protein; - NLCE: 0.176 µg/mg protein; - CSNLCE: 0.307 µg/mg protein	[233]
EGCG	Actively targeted LNs	Mean size: 106 nm Polydispersity index: 0.19 ζ-potential: −20.6 mV Entrapment efficiency: ~95% Loading capacity: ~10%	Incorporation of EGCG into CD36-targeted nanoparticles significantly reduced the secretion of inflammatory cytokines (MCP-1, TNF-α, IL-6) from mouse peritoneal macrophages in vitro and decreased atherosclerotic lesion area by ~30% in LDLr^−/−^ mice, compared to native EGCG and non-targeted formulations.	Encapsulation of EGCG in nanoparticles; Improved EGCG stability; Enhancement of the EGCG bioavailability; Enhancement of EGCG targeted delivery.	[234]
EGCG	Cationic LNs, utilizing dimethyldioctadecylammonium bromide (DDAB) or cetyltrimethylammonium bromide (CTAB)	Mean size: DDAB-based LNs: 143.7 nm CTAB-based LNs: 149.1 nm Polydispersity index: DDAB-based LNs: 0.16 CTAB-based LNs: 0.24 ζ-potential: DDAB-based LNs: +25.7 mV CTAB-based LNs: +20.8 mV Entrapment efficiency: n.i. Loading capacity: n.i.	Cationic EGCG-loaded LNs, formulated with CTAB or DDAB, demonstrated sustained release and effective transcorneal and transscleral permeation, suggesting their strong potential to enhance ocular bioavailability and support the treatment of inflammation- and oxidative stress-related eye disorders.	Encapsulation of EGCG in cationic LNs improved EGCG stability; Increased EGCG bioavailability, when administered onto the ocular mucosa; Enhanced EGCG release and permeation characteristics.	[235]
C	Nanoemulsion (subsequently loaded in Carbopol hydrogel)	Mean size: 98.6 nm Polydispersity index: 0.12 ζ-potential: −27.3 mV Entrapment efficiency: 99.02% Loading capacity: 1.12%	C-loaded nanoemulsion based nano-gel, enhanced transdermal permeation, stability, and antioxidant enzyme restoration in UVA-irradiated skin, showing strong potential for topical anti-inflammatory and photoprotective therapy.	Improved C permeation; Enhancement of C bioavailability: Relative bioavailability of nanoemulsion based nanogel was 894.73% compared to the suspension; C_max_: 93.79 ± 6.19 ng/mL; AUC_0_–∞: 2653.99 ± 515.02 ng·h/mL (over 72 h);	[244]
EGCG	Niosomes	Mean diameter: ~60 nm Polydispersity index: ~0.11 ζ-potential: n.i. Entrapment efficiency: 76.4% Loading capacity: n.i.	Encapsulation of EGCG in Tween 60/cholesterol-based niosomes significantly improved its chemical stability and antioxidant activity under simulated intestinal conditions, increasing residual EGCG from 3% (free EGCG) to 49% after 2 h. Niosomal EGCG showed ~1.5-fold higher cellular antioxidant activity in HepG2 cells compared to free EGCG, even after digestion.	Under simulated intestinal fluid, 49% of EGCG remained after 2 h with niosomes vs. only 3% with free EGCG Niosome-encapsulated EGCG maintained higher activity after intestinal digestion compared to free EGCG	[245]
EGCG	Phytosomes	Mean size: 100 ÷ 250 nm Polydispersity index: n.i. ζ-potential: n.i. Entrapment efficiency: up to 90% Loading capacity: n.i.	EGCG-loaded phytosomes significantly reduced carrageenan-induced paw edema in a rat model compared to both free EGCG and green tea extract. The formulation achieved up to 88.2% inhibition of inflammation after four days, with a noticeable reduction in paw volume observed as early as three hours post-administration. Furthermore, the anti-inflammatory response was sustained throughout the study period, highlighting the prolonged efficacy of the phytosomal system.	Improved EGCG stability and ~90% loading efficiency; Prolonged retention of EGCG; Improved in vivo anti-inflammatory activity in rats (paw edema), compared to pure EGCG and green tea extract.	[246]
C	Niosomes	Mean size: 204.0 nm Polydispersity value: 0.27 ζ-potential: −41.7 mV Entrapment efficiency: 49.5% Loading capacity: n.i.	Incorporation of C into niosomes enhanced dermal delivery and antioxidant effects in UVA-irradiated human fibroblasts. Compared to free C, niosome-encapsulated C increased skin deposition by ~5-fold at 24 h, significantly improved cell viability post-UVA exposure (90% vs. 79%), reduced lipid peroxidation (levels of malondialdehyde—MDA), and enhanced antioxidant enzyme activities (SOD and GSH-Px). Enhanced cellular uptake via energy-dependent endocytosis was also observed.	Enhanced C physicochemical stability; Improved C dermal deposition; Prolonged C release profile; Increased cellular uptake.	[247]
EGCG	Niosomes	Mean size: 235.4 nm Polydispersity index: 0.27 ζ-potential: −45.2 mV Entrapment efficiency: 53.05% Loading capacity: n.i.	Incorporation of EGCG into Span 60-based niosomal nanocarriers significantly enhanced dermal penetration and skin deposition (~2-fold vs. free EGCG) in full-thickness human skin explants. The formulation provided sustained release over 24 h and protected human dermal fibroblasts from UVA-induced oxidative stress by increasing cell viability, reducing intracellular MDA levels, and enhancing antioxidant enzyme activities (SOD, GSH-Px), compared to free EGCG.	Enhanced EGCG physicochemical stability; Skin retention of EGCG was increased by approximately 2 to 3 times in niosomal formulations versus free EGCG in ex vivo skin permeation studies; Prolonged EGCG release profile; Enhanced antioxidant effects.	[248]
C	Hexosomes, with or without sodium taurocholate (ST)	Mean size: With ST: 158.0 nm Without ST: 160.0 nm Polydispersity index: With ST: 0.14 Without ST: 0.13 ζ-potential: With ST: −54.0 mV Without ST: −29.0 mV Entrapment efficiency: With ST: 99.9% Without ST: 99.5% Loading capacity: n.i.	Lipid-based hexosomes incorporating ST and loaded with C significantly enhanced in vitro skin penetration and transdermal delivery by overcoming the stratum corneum barrier in pig skin, while preserving strong antioxidant activity (~88% DPPH inhibition), outperforming non-ST-containing hexosomes and vesicles in terms of penetration depth and cargo loading capacity.	Enhanced physicochemical stability of C; Improved C dermal deposition; Prolonged C release profile; Increased cellular uptake of C.	[249]
Green tea extract	Niosomes	Mean size: ~300 nm Polydispersity index: n.i. ζ-potential: n.i. Entrapment efficiency: n.i. Loading capacity: n.i.	Niosomal green tea extract significantly enhanced cellular antioxidant activity in HepG2 cells compared to native green tea extract and led to greater reductions in plasma total and LDL cholesterol (−25.8% vs. −10.9%) in high-fat-fed C57BL/6 mice. The loaded vesicles upregulated hepatic LDL receptor (+274.7%) and CYP7A1 expression, and downregulated HMG-CoA reductase and SREBP2 mRNA expression more effectively than native green tea extract, indicating improved bioavailability and hypocholesterolemic effects via modulation of cholesterol metabolism and enhanced intracellular antioxidant defense.	Enhanced physicochemical stability; Increased cellular uptake.	[250]

n.i.—no information.

**Table 3 pharmaceutics-17-00985-t003:** Characteristics of various lipid-based nanocarriers for delivering green tea catechins with demonstrated neuroprotective effects in in vitro and/or in vivo studies.

Active Compound/s	Nano-Carrier Type	Nano-Carrier Characteristics	Suggested Mechanism/s of Action	Scientific Result	Reference
C, Quercetin, Fisetin	Liposomes	Mean size: n.i. Polydispersity index: n.i. ζ-potential: n.i. Entrapment efficiency: n.i. Loading capacity: n.i.	Enhancement of in vivo stability; Enhancement of delivery to the brain parenchyma.	The delivery of the bioactive compound to the brain parenchyma in rats (i.p.) was enhanced (brain concentration was 10.5 ng/g for liposomal catechin and undetectable for aqueous form after one i.p. dose)	[251]
C	Liposomes	Mean size: 34.6 ÷ 70.3 nm Polydispersity index: 0.13 ÷ 0.44 ζ-potential: −15.3 ÷ −18.8 mV Entrapment efficiency: 65.8 ÷ 85.6% Loading capacity: n.i.	Protection of the compound from enzymatic degradation; Enhanced bioavailability and delivery of the compound to the brain; Improved stability upon storage.	Suppressed release profile in vitro; Improved in vitro stability under simulated intestinal fluid conditions; Blood levels of liposomal C in rats were elevated at a later stage post-administration; 2.9- and 2.7-fold higher accumulation of C in the cerebral cortex and hippocampus, respectively; Increased concentrations of C in the striatum and thalamus.	[252]
EGCG	Liposomes	Mean size: 132.9 ÷ 161.5 nm Polydispersity index: 0.06 ÷ 0.12 ζ-potential: n.i. Entrapment efficiency: 55.4 ÷ 76.8% Loading capacity: n.i.	Reduction in particle size; Enhancement of the stability of EGCG and liposomes; Increased encapsulation efficiency; Improvement in the bioavailability of EGCG (the cellular uptake of EGCG in microglial cells was enhanced by approximately 3.4 times with the liposomal formulation compared to free EGCG)	Following pre-treatment with EGCG-loaded liposomes, the levels of TNF-α and nitric oxide production in a cellular model of lipopolysaccharide-induced inflammation in BV-2 microglial cells were reduced; Post-treatment with EGCG-loaded liposomes led to symptomatic improvement, suppression of neuroinflammation, and a reduction in TNF-α secretion in a rat model of Parkinson’s disease induced by unilateral injection of lipopolysaccharide into the substantia nigra.	[253]
EGCG	Proliposomal vesicles	Mean size: 150.6 nm Polydispersity index: 0.07 ζ-potential: −71.0 mV Entrapment efficiency: 89.7% Loading capacity: n.i.	Prolonged in vitro release; Enhancement of the EGCG stability.	In a rat model of traumatic brain injury, seven-day pre-treatment with EGCG-proliposomes significantly reduced the lipid peroxidation marker MDA and increased antioxidants (glutathione, SOD). EGCG-proliposomes also more effectively activated the Sirt1/Nrf2/HO-1 pathway; Immunohistochemical analysis showed increased HO-1 expression in the cerebral cortex and hippocampus, further confirmed by histopathological analysis.	[254]
EGCG	Glucose-modified liposomes	Mean size: Glucose-modified EGCG liposomes: 158.7 nm Plain EGCG liposomes: 149.6 nm Polydispersity index: Glucose-modified EGCG liposomes: 0.26 Plain EGCG liposomes: 0.24 ζ-potential: Glucose-modified EGCG liposomes: +2.4 mV Plain EGCG liposomes: +2.1 mV Entrapment efficiency: Glucose-modified EGCG liposomes: 73.1% Plain EGCG liposomes: 71.3% Loading capacity: n.i.	Improved permeability across the BBB and increased neuronal cell uptake, mediated by glucose transporter protein 1 (specifically, the brain concentration of EGCG in the glucose-modified EGCG liposomes group was approximately 2.5 times higher than that in the free EGCG group); Improved stability of both the carriers and the encapsulated EGCG; Improved encapsulation capacity.	Reduced cytotoxicity and enhanced protection against H_2_O_2_-induced oxidative stress; Increased cellular uptake and improved permeability across the BBB via GLUT1 transporters.	[257]
EGCG, Resveratrol	Leptin-modified liposomes	Mean size: 149.8 ÷ 190.4 nm Polydispersity index: n.i. ζ-potential: −18.7 ÷ −41.3 mV Entrapment efficiency: Resveratrol: 54.2% EGCG: 39.5% Loading capacity: n.i.	Targeted delivery of leptin-modified liposomes to the brain.	The surface modification with leptin enabled the liposomes to bind to HBMECs and SH-SY5Y cells via the leptin receptor, enhancing their ability to cross the BBB and be absorbed by the cells; Reductions in the apoptosis-promoting protein Bcl-2-associated X protein and α-synuclein were observed, along with increases in the apoptosis-inhibitory protein B-cell lymphoma 2, tyrosine hydroxylase, and the dopamine transporter.	[258]
EGCG	Nanolipidic particle complexes	Mean size: 30.0 ÷ 80.0 nm Polydispersity index: n.i. ζ-potential: n.i. Entrapment efficiency: n.i. Loading capacity: n.i.	Improved stability and oral bioavailability of EGCG (two-fold higher than that of free EGCG).	Enhanced the neuronal α-secretase activity in vitro by up to 91%; The bioavailability of EGCG was more than two-fold greater after oral administration of the prepared carriers in rats compared to the free form.	[259]
EGCG, Ascorbic acid	Transferosomes	Mean size: 184.3 nm Polydispersity index: 0.18 ζ-potential: −14.2 mV Entrapment efficiency: EGCG: 82.3% Ascorbic acid: 74.8% Loading capacity: EGCG: 12.5% Ascorbic acid: 15.1%	Enhanced stability and bioavailability of EGCG; Increased accumulation of EGCG in major organs, including the brain.	Following intranasal administration in mice, the transferosome-loaded formulation achieved approximately a fivefold increase in long-term brain concentrations of EGCG compared to its free form. In a mouse model of Alzheimer’s disease, intranasal treatment with the transferozomes resulted in enhanced acetylcholinesterase activity, reduced neuroinflammation, and improvements in spatial learning and memory.	[260]
EGCG	SLNs	Mean size: 162.4 nm Polydispersity index: n.i. ζ-potential: n.i. Entrapment efficiency: n.i. Loading capacity: n.i.	Enhanced EGCG bioavailability (two-fold greater than that of free EGCG); Improved delivery of EGCG to the brain.	Improved brain bioavailability of EGCG; Marked alleviation of memory deficits in a rodent model of cerebral ischemia.	[261]

n.i.—no information.

**Table 4 pharmaceutics-17-00985-t004:** Characteristics of various lipid-based nanocarriers for delivering green tea catechins with demonstrated anticarcinogenic activity in in vitro and/or in vivo studies.

Active Compound/s	Nano-Carrier Type	Nano-Carrier Characteristics	Suggested Mechanism/s of Action	Scientific Result	Reference
EGCG	SLNs	Mean size: 300.2 nm Polydispersity index: 0.418 ζ potential: −18.0 mV Entrapment efficiency: 81.0% Loading capacity: n.i.	Pharmacokinetic studies in Wistar rats demonstrated higher plasma concentrations of EGCG compared to its free form; The oral bioavailability of EGCG in SLNs was observed to be more than two-fold greater than that of free EGCG; Histopathological and toxicological evaluations confirmed the absence of treatment-related adverse effects.	Improved stability and bioavailability of EGCG upon formulation into SLNs; EGCG-loaded SLNs were demonstrated to be an efficient system for the sustained release of EGCG.	[267]
C, EC, EGCG	Liposomes	Mean size: 131.1 ÷ 378.2 nm Polydispersity index: n.i. ζ potential: −36.1 ÷ −0.9 mV Entrapment efficiency: C: 39.5 ÷ 57.0% EC: 31.9 ÷ 64.7% EGCG: 84.6 ÷ 99.6% Loading capacity: n.i.	Significantly increased EGCG tumor uptake, as well as intratumoral retention and distribution (EGCG deposition was increased by 20-fold with liposomal preparation, while free EGCG showed virtually no detectable accumulation in basal cell carcinomas); Superior cytotoxicity toward basal cell carcinoma cells than hydroalcoholic solutions, even at lower EGCG doses; Potential for dose reduction and fewer side effects.	Improved stability of EGCG; Liposomal formulations incorporating deoxycholic acid and 15% ethanol substantially enhanced the intratumoral accumulation of EGCG; The surface charge and bilayer fluidity of the liposomes were identified as critical determinants governing their distribution within tumor tissue. Additionally, increased vesicle size was shown to favor prolonged retention in the tumor microenvironment, likely due to restricted vascular clearance and entrapment within the extracellular matrix.	[274]
EGCG	SLNs, conjugated with bombesin	Mean size: 163.4 nm Polydispersity index: 0.34 ζ potential: −25.2 mV Entrapment efficiency: 67.2% Loading capacity: n.i.	Enhanced stability; Elevated liposome internalization and EGCG accumulation in MCF7 cells; Improved survival and reduced tumor volume in C57BL/6 mice.	Enhanced bioavailability, solubility, physicochemical stability, and encapsulation capacity of EGCG; Protection of EGCG from degradation under physiological pH conditions while concurrently improving its cytotoxic efficacy and cellular uptake.	[275]
EGCG	Cationic SLNs	Mean size: ~143.7 nm Polydispersity index: 0.16 ζ potential: +25.7 mV Entrapment efficiency: 96.9% Loading capacity: ~15%	Slightly enhanced antiproliferative effect in MCF-7 and SV-80 cells; Inherent toxicity of SLNs, attributed to the surfactant utilized.	Improved stability and protection of EGCG from degradation; Increased cytotoxicity, improved cellular uptake, resulting in better therapeutic efficacy.	[276]
EGCG	Chitosan-coated liposomes	Mean size: 85.0 nm Polydispersity index: 0.35 ζ potential: +16.4 mV Entrapment efficiency: ~90% Loading capacity: ~3%	Enhanced EGCG stability and prolonged sustained release profile; Increased intracellular accumulation of EGCG in MCF7 breast cancer cells; Augmented induction of apoptosis and inhibition of MCF7 cell proliferation compared to free EGCG (the chitosan-coated liposomes maintained antiproliferative and proapoptotic effects even at concentrations as low as 10 µM, where native EGCG shows no significant activity).	The encapsulation protected EGCG from degradation and enabled controlled release; Chitosan coating enhanced cellular uptake by promoting receptor interactions, improving bioavailability; The liposomal structure sustained EGCG release, prolonging therapeutic effect and anticancer activity.	[277]
C, EC, EGCG	Liposomes	Mean size: 135.2 ÷ 268.9 nm Polydispersity index: n.i. ζ potential: −66.0 ÷ +68.6 mV Entrapment efficiency: C: 41.9 ÷ 75.6% EC: 37.3 ÷ 76.9% EGCG: 36.3 ÷ 89.7% Loading capacity: n.i.	Compared to aqueous solutions, liposomal formulations delivered substantially higher amounts of catechins to solid tumors (intratumor deposition of liposomal EGCG with deoxycholic acid and ethanol was 20-fold higher compared to free aqueous EGCG); No significant increase in catechins deposition within the skin was observed following topical liposomal application.	The incorporation of anionic agents—specifically deoxycholic acid or dicetyl phosphate—enhanced the permeability of the lipid bilayer membranes, thereby facilitating a more rapid release of the encapsulated compounds; Enhanced drug accumulation within tumor tissues was achieved through the optimization of both the vesicle size and the drug release kinetics.	[278]
C, EGCG	Niosomes	Mean size: ~100 nm Polydispersity index: n.i. ζ potential: n.i. Entrapment efficiency: above 40% Loading capacity: n.i.	Cellular uptake of both C and EGCG was significantly enhanced in Caco-2 cells; Niosomal formulations demonstrated 1.8-fold (C) and 1.4-fold (EGCG) increases in transcellular flux compared to their free counterparts; Encapsulated forms exhibited higher IC_50_ values, indicating reduced cytotoxicity alongside greater transport efficiency, thereby necessitating dose recalibration in relation to encapsulation efficiency.	C and EGCG exhibited encapsulation efficiencies exceeding 40%; The nanoscale vesicular architecture facilitated uptake via endocytosis; Encapsulation conferred protection against metabolic degradation; Niosomal formulations circumvented efflux mediated by P-glycoprotein and multidrug resistance-associated protein 2; Enhanced permeability was attributed to surfactant-induced membrane interactions and EDTA-mediated loosening of tight junctions.	[279]
EGCG	SLNs	Mean size: 112.5 ÷ 157.4 nm Polydispersity index: 0.13 ÷ 0.27 ζ potential: −30.1 ÷ −37.2 mV Entrapment efficiency: 67.2 ÷ 89.5% Loading capacity: n.i.	The in vitro drug release profile demonstrated an initial rapid release of approximately 10% within the first hour, followed by a sustained release reaching 83.9% by 12 h; In vitro cytotoxicity assays indicated that blank SLNs exhibited high biocompatibility, maintaining over 90% cell viability; EGCG-loaded SLNs caused a significant, dose-dependent reduction in viability of two cancer cell lines: MDA-MB-231 (breast cancer) and DU-145 (prostate cancer); EGCG-loaded SLNs induced markedly greater apoptosis in cancer cells compared to free EGCG treatment.	EGCG exhibited enhanced stability within the formulation; Controlled drug release is likely mediated through diffusion or erosion of the lipid matrix; Increased cytotoxicity is attributed to improved cellular uptake and selective tumor targeting, facilitated by the enhanced permeation and retention effect that promotes nanoparticle accumulation in tumor tissues.	[280]
C	Liposomes	Mean size: 221.0 nm Polydispersity index: n.i. ζ potential: n.i. Entrapment efficiency: 76% Loading capacity: n.i.	Demonstrated high stability and efficient encapsulation efficiency; Superior antioxidant performance over free C, with enhanced scavenging of ABTS, hydroxyl, and DPPH radicals in vitro; Exhibited cytotoxicity toward Caco-2 cells only at higher concentrations (≥0.025 mg/mL), with increased toxicity observed at 36 h compared to 24 h.	Under the optimized conditions, C was successfully loaded into liposomes, resulting in a stable nanocarrier system with uniform size and well-defined morphology.	[281]
Paclitaxel, EGCG	Liposomes	Mean size: 130.5 nm Polydispersity index: 0.42 ζ potential: −36.77 mV Entrapment efficiency: Paclitaxel: 77.1% EGCG: 59.1% Loading capacity: n.i.	Liposomes successfully co-encapsulated Paclitaxel and EGCG despite their differing solubilities; Paclitaxel-EGCG-loaded liposomes at a 1:5 molar ratio exhibited significantly enhanced cytotoxicity compared to the free drugs or their respective single-drug liposomal formulations; This combination induced a greater extent of apoptosis, with caspase-3 activity increasing approximately 3.9-fold relative to the vehicle control; Treatment with Paclitaxel/EGCG liposomes resulted in a marked reduction (~80%) in the activity of matrix metalloproteinases MMP-2 and MMP-9; Paclitaxel/EGCG liposomes also produced the most pronounced inhibition of tumor cell invasion among all tested formulations.	The phospholipid bilayer structure of the liposomes facilitated both the efficient co-encapsulation and intracellular delivery of both Paclitaxel and EGCG; The co-loaded liposomes demonstrated synergistic anticancer effects by simultaneously promoting apoptosis, suppressing MMP activity, and limiting tumor cell invasiveness.	[282]
C	C-functionalized cationic lipopolymer based multicomponent nanomicelles	Mean size: n.i. Polydispersity index: n.i. ζ-potential: n.i. Entrapment efficiency: n.i. Loading capacity: n.i	Improved intracellular delivery and bioavailability of C; Targeted accumulation in pulmonary tissue, rendering the system highly suitable for the treatment of lung-associated diseases; Prolonged systemic circulation of C, thereby enhancing the therapeutic window; Selective cytotoxicity toward cancer cells, validating the chemopreventive efficacy of the delivery system.	Electrostatic interactions between cationic carriers and cell membranes enhance lung-targeted intracellular uptake; Serum albumin forms stable complexes with lipopolymer, reducing clearance and prolonging circulation; Cationic liposomes accumulate preferentially in pulmonary microvasculature due to surface charge; Functionalized nanomicelles induce ROS production and caspase-3 activation, promoting cancer cell apoptosis.	[283]

n.i.—no information.

**Table 5 pharmaceutics-17-00985-t005:** Characteristics of various lipid-based nanocarriers for delivering green tea catechins with demonstrated antimicrobial activity in in vitro and/or in vivo studies.

Active Compound/s	Nano-Carrier Type	Nano-Carrier Characteristics	Suggested Mechanism/s of Action	Scientific Result	Reference
EGCG	Liposomes	Mean size: neutral: 93.2 nm cationic: 93.4 nm anionic: 89.4 nm Polydispersity index: neutral: 0.22 cationic: 0.22 anionic: 0.23 ζ-potential: neutral: −1.2 mV cationic: +18.7 mV anionic: −13.2 mV Entrapment efficiency: neutral: 56% cationic: 78% anionic: 45% Loading capacity: n.i.	An electrostatic interaction between the outer membrane of MRSA and the cationic liposomes may enhance EGCG entry into the bacterial cell; Minimum inhibitory concentrations: Free EGCG: 128 mg/L Cationic liposomes: 16 mg/L Neutral liposomes: 32 mg/L Anionic liposomes: 256 mg/L	The cationic and neutral EGCG-loaded liposomes are more potent than free EGCG; The cationic EGCG-loaded liposomes led to 100% survival rate in mice (no mice survived in the control group). Survival rates in burned mouse skin infected by MRSA: - Cationic nanoliposomes: ~100% survival - Neutral nanoliposomes: ~70% survival - Anionic nanoliposomes: ~30% survival	[286]
EGCG	Lipid-chitosan hybrid nanoparticles	Mean size: 217.3 nm Polydispersity index: 0.17 ζ-potential: +33.7 mV Entrapment efficiency: 95.9% Loading capacity: n.i.	Improved antimicrobial activity of EGCG; Mucoadhesive properties of the nanoparticles due to chitosan content that conferred a positive ζ-potential leading to an interaction between the nanoparticle and mucin	The EGCG-loaded lipid-chitosan hybrid nanoparticles exert higher antibacterial activity compared to free EGCG and lipid-chitosan hybrid nanoparticles alone (reduction of ∼15-fold in the MIC and MBC against *Streptococcus mutans*, *Streptococcus sobrinus*, and *Lactobacillus casei*, compared to free EGCG).	[287]
Tea polyphenols	Liposomes	Mean size: 66.8 nm Polydispersity index: 0.21 ζ-potential: −6.16 mV Entrapment efficiency: 78.5% Loading capacity: 8.5%	High viscidity of the lipid layer may reduce the diffusion of tea polyphenols into the agar.	The antibacterial activity of GTCs was reduced when they are encapsulated in liposomes.	[288]
Green tea extract	SLNs	Mean size: 175 ÷ 275 nm Polydispersity index: 0.20 ÷ 0.28 ζ-potential: −34.1 ÷ −46.6 mV Entrapment efficiency: n.i. Loading capacity: n.i.	n.i.	Green tea extract-loaded SLNs showed a good physical stability and exhibited antioxidant and antibacterial activity.	[289]
C	C-in-cyclodextrine-in-liposomes	Mean size: 415.0 nm Polydispersity index: 0.54 ζ-potential: −40.6 mV Entrapment efficiency: 98.9% Loading capacity: n.i.	n.i.	C-in-cyclodextrin-in-phospholipid liposomes showed higher water solubility (25.13%) and in vitro permeability (42.14%) compared to C as well as higher antibacterial activity.	[290]

n.i.—no information.

## Data Availability

The data presented in this study are available on request from the corresponding author.

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
