# Peer review of "Lipid-Based Nanotechnologies for Delivery of Green Tea Catechins: Advances, Challenges, and Therapeutic Potential"

_pharmaceutics, 2025, doi:10.3390/pharmaceutics17080985_

Round 1

Reviewer 1 Report

Comments and Suggestions for Authors

Review of Manuscript entitled: «Lipid-Based Nanotechnologies for Delivery of Green Tea Catechins: Advances, Challenges, and Therapeutic Potential »

I found that the material in the article is quite interesting and I appreciate the idea and works have done. I would like to thank the authors for their careful work. Please find below my comment:

1) It would be interesting to examine the content of beneficial catechins not only in teas with varying degrees of fermentation but also, more importantly, in relation to their geographic origin. I recommend that the authors consider this aspect. It is possible that the authors may identify a specific tea variety and optimal fermentation level that yield the best product. However, what will the authors do if these findings are only applicable to regions such as high-altitude areas in China, for example?

2) In addition to the previous question, how does soil affect tea?

3) I have a question regarding the use of tea as a source of catechins. Is it not simpler to purchase pure reagents instead? What is the appeal of using tea specifically?

4) The authors state: "To a great extent, the in vitro and in vivo behavior of catechins in all aspects (e.g., solubility, extraction efficiency, stability, complexation, absorption, biotransformation, antioxidant activity, etc.) is determined by the number and positioning of the hydroxyl groups in their structure (counting 5 to 8), and more specifically—by the number and strength of intra- and intermolecular hydrogen bonds occurring." These statements require further clarification. I understand that the authors have provided references to sources, but as they mention, low stability is a key factor, and it is essential to investigate this matter in greater depth.

5) It would be very helpful if the authors could create a comprehensive table summarizing the pharmacological effects. In this table, it would be beneficial to indicate which effects have been studied in vivo and which in vitro. This would facilitate a more realistic assessment of the current usefulness of catechins.

6) I always have one question regarding nanoparticle-based carriers: What is their stability within the body? Nano-objects tend to aggregate into larger particles in aqueous environments. Given the effect of the protein corona in the body, the use of nano-objects for drug delivery appears to be an ineffective idea at this time. What do the authors think about this?

7) Regarding neuroprotective efficacy, the authors should exercise caution on this topic. Thousands of researchers are struggling with the challenge of overcoming the blood-brain barrier to deliver drugs to the brain. The authors should be critical of this issue and conduct a more thorough examination of the sources. Additionally, it would be very helpful for the review if the authors could visualize the process of such delivery.

8) A general comment about the article: there are no illustrations. For a general audience, the lack of illustrative material makes the article difficult to comprehend.

9) I am interested in how the issue purity of nanoparticles derived from biological sources is being investigated.

I recommend the publication of this work after major revision in Pharmaceutics.

Author Response

Dear Reviewer 1,

We sincerely appreciate the opportunity to address your comments, as they provide valuable insights and help us enhance the quality of our work.

Here are the responses needed:

Comments 1: It would be interesting to examine the content of beneficial catechins not only in teas with varying degrees of fermentation but also, more importantly, in relation to their geographic origin. I recommend that the authors consider this aspect. It is possible that the authors may identify a specific tea variety and optimal fermentation level that yield the best product. However, what will the authors do if these findings are only applicable to regions such as high-altitude areas in China, for example?

Response 1: The manuscript has been updated to address the comment.

The manuscript has been updated to address the comment (Lines 157–162).

Comments 2: In addition to the previous question, how does soil affect tea?

Response 2: The manuscript has been updated to address the comment (Lines 162–166).

Comments 3: I have a question regarding the use of tea as a source of catechins. Is it not simpler to purchase pure reagents instead? What is the appeal of using tea specifically?

Response 3: The manuscript has been updated to address the comment (Lines 226–244).

Comments 4: The authors state: "To a great extent, the in vitro and in vivo behavior of catechins in all aspects (e.g., solubility, extraction efficiency, stability, complexation, absorption, biotransformation, antioxidant activity, etc.) is determined by the number and positioning of the hydroxyl groups in their structure (counting 5 to 8), and more specifically—by the number and strength of intra- and intermolecular hydrogen bonds occurring." These statements require further clarification. I understand that the authors have provided references to sources, but as they mention, low stability is a key factor, and it is essential to investigate this matter in greater depth.

Response 4: Thank you for this remark. Indeed, as we also believe, stability is the most crucial issue when it comes to utilizing green tea catechins for medicinal purposes. It is also the most important obstacle preventing their mass use and clinical introduction. This matter is briefly regarded in section 3.3, but mostly, we would like to direct your attention to the two paragraphs following Table 1, Lines 259-268 and 269-286, respectively.

Comments 5: It would be very helpful if the authors could create a comprehensive table summarizing the pharmacological effects. In this table, it would be beneficial to indicate which effects have been studied in vivo and which in vitro. This would facilitate a more realistic assessment of the current usefulness of catechins.

Response 5: All pharmacological effects discussed have been demonstrated both in vitro and in vivo, with supporting evidence presented in tabular format in Sections 6 through 9 inclusive.

Comments 6: I always have one question regarding nanoparticle-based carriers: What is their stability within the body? Nano-objects tend to aggregate into larger particles in aqueous environments. Given the effect of the protein corona in the body, the use of nano-objects for drug delivery appears to be an ineffective idea at this time. What do the authors think about this?

Response 6: The concerns about the stability of nanocarriers in the body, particularly their tendency to aggregate in aqueous media, as well as the protein corona impact, are well-founded. While nanoparticles may be prone to aggregation due to hydrophobic interactions or charge neutralization in physiological environments, the presence and composition of the biomolecular corona can both stabilize and destabilize nanoplatforms depending on protein type and concentration. Plasma proteins, such as ApoA1 and serum albumin, are shown to stabilize gold nanoparticles by forming an effective coating, thereby preventing interparticle interactions. Conversely, larger molecules such as fibrinogen and immunoglobulins can promote cross-linking between the particles. We should not forget the researchers’ attempts to protect nanocarriers by creating a corona in advance (entropic shielding), so they can be engineered to achieve more predictable targeting or to avoid rapid clearance by the reticuloendothelial system. In a nutshell, the presence of protein corona does not render nanocarriers inherently ineffective. Indisputably, the research in this direction should be broadened, especially towards maximal in vitro replication of in vivo corona formation.

Comments 7: Regarding neuroprotective efficacy, the authors should exercise caution on this topic. Thousands of researchers are struggling with the challenge of overcoming the blood-brain barrier to deliver drugs to the brain. The authors should be critical of this issue and conduct a more thorough examination of the sources. Additionally, it would be very helpful for the review if the authors could visualize the process of such delivery.

Response 7: The manuscript has been revised to include a response to the comment, as well as a figure illustrating the translocation of GTCs across the blood–brain barrier (Lines 353–380).

Comments 8: A general comment about the article: there are no illustrations. For a general audience, the lack of illustrative material makes the article difficult to comprehend.

Response 8: A revision has been made to enhance the illustrative quality of the manuscript.

Comments 9: I am interested in how the issue purity of nanoparticles derived from biological sources is being investigated.

Response 9: To ensure the safe use of biologically-derived nanoparticles, their purity is ensured by purification steps and investigated by a combination of physicochemical methods, biochemical assays, and biological protocols. The current regulatory frameworks apply existing drug and biologics regulations on a case-by-case basis. However, clear guidance on nanocarriers remains under development.

Reviewer 2 Report

Comments and Suggestions for Authors

Review report

I have gone through the manuscript titled “Lipid-based nanotechnologies for delivery of green tea cate-chins: advances, challenges, and therapeutic potential” by Stoeva-Grigorova et al.

The manuscript offers an in-depth review of lipid-based nanotechnologies for delivering green tea catechins (GTCs). It is scientifically rigorous, with abundant references and technical details about the physicochemical properties of catechins, bioavailability challenges, and potential solutions via nanoencapsulation. It presents valuable information of current literature and could serve as a reference for researchers in pharmaceutical sciences, nanotechnology, phytochemistry, and nutraceuticals.

The review is in-depth and of importance to the researchers working in this field. However, the manuscript needs to be strengthened with respect to the following points before reconsidering for publication. Hence, a major review is recommended at this stage.

  • Among lipid-based nanocarriers, which ones have demonstrated the highest bioavailability improvements for EGCG or other catechins in vivo? Could the authors provide a clearer comparison?
  • What is the intended primary audience of this review-pharmaceutical scientists, clinicians, food technologists, or nanotechnology researchers? Consider clarifying this in the introduction.
  • Are there any human clinical trials utilizing lipid-based nanocarriers for GTC delivery? If so, results should be explicitly summarized.
  • Could the authors expand on toxicity or safety concerns specific to these nano-formulations, especially cationic nanoparticles and surfactants like CTAB or DDAB?
  • What is the current regulatory status of lipid-based nanocarriers for catechin delivery?
  • How feasible is the large-scale production of these nanocarriers for pharmaceutical or nutraceutical industries? What are the main bottlenecks?
  • How does the nanoparticle size specifically influence catechin release kinetics and cellular uptake? Some discussion with figures/tables might help to improve the manuscript.
  • On what chemical or physical basis should researchers choose between nanoemulsions, liposomes, SLNs, NLCs, etc., for catechins?
  • The review mentions pro-oxidant behavior of GTCs. How do lipid carriers influence this dual antioxidant/pro-oxidant activity?
  • The review presents many in vitro /ex vivo Could the authors explicitly discuss discrepancies between in vitro and in vivo performance?
  • The manuscript is long and technical. Would the authors consider adding a graphical abstract summarizing the key findings? The authors may also consider including diagrams, flowcharts throughout the manuscript.
  • The authors may consider adding a dedicated subsection discussing regulatory pathways, approvals, and market status for such nanocarriers.
  • The authors may consider including summary tables comparing pharmacokinetic parameters of free vs nano-encapsulated catechins.
  • The authors could add a dedicated subsection discussing regulatory pathways, approvals, and market status for such nanocarriers.
  • What is the significance of the "antioxidant-to-prooxidant transitioning behavior" of GTCs mentioned in the abstract? Could the authors elaborate on the conditions under which this transition occurs and its potential physiological consequences?
  • The "Methods" section states that the literature search retrieved "over 270 relevant sources". Could the authors provide more details on the specific search terms used and the inclusion/exclusion criteria applied to ensure reproducibility?
  • The review mentions that cationic EGCG-loaded nanoliposomes had a much lower minimum inhibitory concentration (MIC) against MRSA compared to anionic nanoliposomes and free EGCG. What is the proposed mechanism behind the enhanced antimicrobial activity of the cationic formulation?
  • Table 1 provides physicochemical data for some GTCs, but many entries are missing solubility data, such as for EC, EGC, and ECG. Could the authors include this missing data to make the table more complete and useful for readers? Also, the authors may consider adding a column for the number of hydroxyl groups to directly link structure to properties.
  • A summary table in the "Pharmacological Activity" section that compares the different lipid-based systems (e.g., liposomes, solid lipid nanoparticles, nanostructured lipid carriers) based on their advantages, disadvantages, and application for GTC delivery will be highly beneficial. This would help the reader compare the different technologies easily.
  • While the review touches on challenges, a dedicated and more in-depth section could be added to discuss manufacturing scalability, stability during storage, and in vivo toxicity concerns.

Author Response

Dear Reviewer 2,

We truly value the opportunity to respond to your comments, as they offer important perspectives that enable us to improve the quality of our work.

Here are the responses needed:

Comments 1: Among lipid-based nanocarriers, which ones have demonstrated the highest bioavailability improvements for EGCG or other catechins in vivo? Could the authors provide a clearer comparison?

Response 1: In most of the included in vivo studies, the authors assess certain effects (e.g., anti-inflammatory potential, effects on plasma lipid profile and traumatic brain injury, antiparkinsonian and anti-Alzheimer activity, anticarcinogenic potential, antibacterial effects, etc.). Only a few of the studies are investigating the bioavailability of the corresponding catechin. Harwansh et al. found increased bioavailability upon transdermal delivery of (+)-catechin-containing nanogel (compared to an orally administered suspension thereof). Moreover, Rivera et al. found increased cerebral concentrations of the same representative after intraperitoneal application of liposomes containing it compared to aqueous solution. Additionally, orally applied elastic liposomes resulted in a significant increase in blood and brain (+)-catechin levels compared to oral aqueous solution, as observed in the study by Y.-B. Huang et al. Smith et al. have found that nanolipidic particle complexes provided enhanced EGCG bioavailability after oral administration. Enhanced organ distribution (including in the brain) after transferosomes-loaded EGCG nasal application was established by Mishra et al. Oral administration of SLN-EGCG provided a significant enhancement in plasma concentrations in the study of Ramesh et al. Overall, we believe these data are insufficient to make a meaningful comparison.

Comments 2: What is the intended primary audience of this review-pharmaceutical scientists, clinicians, food technologists, or nanotechnology researchers? Consider clarifying this in the introduction.

Response 2: The manuscript has been updated to address the comment (Lines 101–106).

Comments 3: Are there any human clinical trials utilizing lipid-based nanocarriers for GTC delivery? If so, results should be explicitly summarized.

Response 3: The manuscript has been updated to address the comment (Lines 1040–1045).

Comments 4: Could the authors expand on toxicity or safety concerns specific to these nano-formulations, especially cationic nanoparticles and surfactants like CTAB or DDAB?

Response 4: The manuscript has been updated to address the comment (Lines 730–741).

Comments 5: What is the current regulatory status of lipid-based nanocarriers for catechin delivery?

Response 5: The manuscript has been updated to address the comment (Lines 664–673).

Comments 6: How feasible is the large-scale production of these nanocarriers for pharmaceutical or nutraceutical industries? What are the main bottlenecks?

Response 6: The concerns about the large-scale production of the lipid-based nanocarriers in question have been mentioned in Section 5. Regarding the main limitations, unfortunately, they are not a few. Nanocarriers’ fundamental properties (particle size, dispersity, morphology, surface characteristics, and entrapment efficiency) can be deteriorated during the transition from laboratory to large-scale production. Moreover, often unattainable remains the achievement of consistent batch-to-batch reproducibility, especially for complex formulations involving multiple components or surface modifications. High production costs and low yields affect economic viability and product safety. Some manufacturing techniques can be harsh for thermolabile drugs or biomolecules, causing chemical instability or degradation during production. Nevertheless, the attempts to scale up the production of these nanocarriers are ongoing.

Comments 7: How does the nanoparticle size specifically influence catechin release kinetics and cellular uptake? Some discussion with figures/tables might help to improve the manuscript.

Response 7: The studies included in this review, regardless of the type of lipid nanocarriers and their size, all state that they assessed modified drug release and enhanced cellular uptake, and these results are presented in Tables 2–5. The researchers employ different types of carriers, although they are all lipid-based. Also, they investigate the uptake of various catechin representatives in different types of cells. How should we make a comparison based only on size differences? We believe it would be inappropriate to make such conclusions. We can only make the well-known but redundant general statements—nanocarriers with dimensions below 100 nm are thought to possess better ability to penetrate across biological membranes, thereby benefiting cellular uptake; smaller nanoplatforms have higher total surface area-to-volume ratio and can facilitate greater and more rapid release.

Comments 8: On what chemical or physical basis should researchers choose between nanoemulsions, liposomes, SLNs, NLCs, etc., for catechins?

Response 8: Each of the discussed nanocarriers possesses the ability to encapsulate BACs with variable physicochemical characteristics, while stabilizing them and enhancing their permeability and bioavailability. These lipid-based nanoplatforms can be designed using various structural components and by applying different production methods. It is common knowledge that pre-formulation studies are crucial regarding the encapsulation efficiency, stability, and bioaccessibility. By carefully selecting excipients and processing parameters, researchers can tailor the choice to align with their specific interests.

Comments 9: The review mentions pro-oxidant behavior of GTCs. How do lipid carriers influence this dual antioxidant/pro-oxidant activity?

Response 9: The manuscript has been updated to address the comment (Lines 786–798).

Comments 10: The review presents many in vitro /ex vivo. Could the authors explicitly discuss discrepancies between in vitro and in vivo performance?

Response 10: The manuscript has been updated to address the comment (Lines 1059–1073).

Comments 11: The manuscript is long and technical. Would the authors consider adding a graphical abstract summarizing the key findings? The authors may also consider including diagrams, flowcharts throughout the manuscript.

Response 11: Thank you for your suggestions and the opportunity to extant the graphical representation of our paper. A graphical abstract was created and added to the manuscript to highlight the main findings.

Comments 12: The authors may consider adding a dedicated subsection discussing regulatory pathways, approvals, and market status for such nanocarriers.

Response 12: In response to comment 5, Section 5 was expanded. Thus, it also responds to the present comment (Lines 664–673).

Comments 13: The authors may consider including summary tables comparing pharmacokinetic parameters of free vs nano-encapsulated catechins.

Response 13: Since not all authors have conducted equally comprehensive pharmacokinetic studies, comparative data regarding the pharmacokinetic parameters of free versus nano-encapsulated catechins have been incorporated into the respective tables within Sections 6 through 9, inclusive.

Comments 14: The authors could add a dedicated subsection discussing regulatory pathways, approvals, and market status for such nanocarriers.

Response 14: We would kindly like to point out that Comment No. 12 appears to be a duplicate of Comment No. 14.

Comments 15: What is the significance of the "antioxidant-to-prooxidant transitioning behavior" of GTCs mentioned in the abstract? Could the authors elaborate on the conditions under which this transition occurs and its potential physiological consequences?

Response 15: The manuscript has been updated to address the comment (Lines 406–411).

Comments 16: The "Methods" section states that the literature search retrieved "over 270 relevant sources". Could the authors provide more details on the specific search terms used and the inclusion/exclusion criteria applied to ensure reproducibility?

Response 16: Both inclusion and exclusion criteria have been incorporated into the manuscript (Lines 116–141).

Comments 17: The review mentions that cationic EGCG-loaded nanoliposomes had a much lower minimum inhibitory concentration (MIC) against MRSA compared to anionic nanoliposomes and free EGCG. What is the proposed mechanism behind the enhanced antimicrobial activity of the cationic formulation?

Response 17: The manuscript has been updated to address the comment (Lines 998–1002).

Comments 18: Table 1 provides physicochemical data for some GTCs, but many entries are missing solubility data, such as for EC, EGC, and ECG. Could the authors include this missing data to make the table more complete and useful for readers? Also, the authors may consider adding a column for the number of hydroxyl groups to directly link structure to properties.

Response 18: Despite our best efforts, we were not able to find and provide the missing experimental data among research reports. Table 1 was upgraded according to your recommendation (Line 255).

Comments 19: A summary table in the "Pharmacological Activity" section that compares the different lipid-based systems (e.g., liposomes, solid lipid nanoparticles, nanostructured lipid carriers) based on their advantages, disadvantages, and application for GTC delivery will be highly beneficial. This would help the reader compare the different technologies easily.

Response 19: The section titled "Pharmacological Activity" aims to provide a comprehensive summary of the established biological effects of GTCs, irrespective of whether they are administered in free form or encapsulated. Consequently, such comparative tables have been created but are presented individually within each of the sections from number 6 through 9, inclusively.

Comments 20: While the review touches on challenges, a dedicated and more in-depth section could be added to discuss manufacturing scalability, stability during storage, and in vivo toxicity concerns.

Response 20: Concerns regarding in vivo toxicity were discussed in Section 7, which also included the daily doses associated with liver damage as reported by various health authorities. The scalability and storage stability of the lipid-based nanocarriers is generally discussed in Section 5.

Round 2

Reviewer 1 Report

Comments and Suggestions for Authors

I recommend accept in present form this paper.

Reviewer 2 Report

Comments and Suggestions for Authors

The authors have provided suitable responses to the review comments and have modified the manuscript suitably. The manuscript is now suitable for publication.